# Unlocking plant health survey data: An approach to quantify the sensitivity and specificity of visual inspections

Matt Combes[1,2]*, Nathan Brown[3,4], Robin N. Thompson[5], Alexander Mastin[6], Peter Crow[4], Stephen Parnell[1,2]

**1** Warwick Crop Centre, University of Warwick, Stratford-upon-Avon, United Kingdom, **2** Zeeman Institute for Systems Biology and Infectious Disease Epidemiology Research, University of Warwick, Coventry, United Kingdom, **3** Woodland Heritage, Haslemere, United Kingdom, **4** Forest Research, Farnham, United Kingdom, **5** Mathematical Institute, University of Oxford, Oxford, United Kingdom, **6** Epidemiology & Risk Policy Advice, Animal and Plant Health Agency, London, United Kingdom

* matt.combes@warwick.ac.uk

## Abstract

Invasive plant pests and pathogens cause substantial environmental and economic damage. Visual inspection remains a central tenet of plant health surveys, but its sensitivity (probability of correctly identifying the presence of a pest) and specificity (probability of correctly identifying the absence of a pest) are not routinely quantified. As knowing sensitivity and specificity of visual inspection is critical for effective contingency planning and outbreak management, we address this deficiency using empirical data and statistical analyses. Twenty-three citizen scientist surveyors assessed up to 175 labelled oak trees for three symptoms of acute oak decline. The same trees were also assessed by an expert who has monitored these individual trees annually for over a decade. The sensitivity and specificity of surveyors was calculated using the expert data as the 'gold-standard' (i.e., assuming perfect sensitivity and specificity). The utility of an approach using Bayesian modelling to estimate the sensitivity and specificity of visual inspection in the absence of a rarely available 'gold-standard' dataset was then examined with simulated plant health survey datasets. There was large variation in sensitivity and specificity between surveyors and between different symptoms, although the sensitivity of detecting a symptom was positively related to the frequency of the symptom on a tree. By leveraging surveyor observations of two symptoms from a minimum of 80 trees on two sites, with reliable prior knowledge of sites with a higher (~0.6) and lower (~0.3) true disease prevalence we show that sensitivity and specificity can be estimated without 'gold-standard' data using Bayesian modelling. We highlight that sensitivity and specificity will depend on the symptoms of a pest or disease, the individual surveyor, and the survey protocol. This has consequences for how surveys are designed to detect and monitor outbreaks, as well as the interpretation of survey data that is used to inform outbreak management.

**Data availability statement:** The code and data used for this paper are available from: https://github.com/MCombess/Plant_Health_sens_spec_workflow and are archived: https://doi.org/10.5281/zenodo.15730414.

**Funding:** MC, NB, PC, SP undertook the work with funding from the United Kingdom's Department for Environment, Food & Rural Affairs (https://www.gov.uk/government/organisations/department-for-environment-food-rural-affairs) through the Future Proofing Plant Health Programme (Project Reference: TH42222FR09: citizen science sensitivity and specificity). MC, SP, NB also received funding from United Kingdom's Research and Innovation (UKRI) Programme through the Natural Environment Research Council (NERC) (https://www.ukri.org/councils/nerc/) (Project: NE/T007729/1). The funders had no role in study design, data collection and analysis, decision to publish, or preparation of the manuscript.

**Competing interests:** The authors have declared that no competing interests exist.

## Author summary

The increasing occurrence of emerging plant pests and diseases is affecting both agricultural and natural ecosystems. Effective management and control of such pests and diseases is much easier when they are detected early. Currently, visual surveys underpin plant health surveillance, but basic metrics of the reliability of visual detection such as the sensitivity (probability of correctly identifying a positive) and specificity (probability of correctly identifying a negative) are not routinely quantified. In this study, we first quantify the sensitivity and specificity of 23 trained citizen scientist surveyors at detecting three symptoms of acute oak decline, by comparing their symptom classifications against a dataset from an expert who has conducted long-term monitoring of these individual trees. We demonstrate how individuals vary greatly in their ability to detect symptoms, and how different symptoms are associated with different detection error. Secondly, based on this dataset we outline an approach developed for scenarios realistic in plant health which utilises Bayesian modelling to estimate the sensitivity and specificity in the absence of a rarely available 'gold-standard' (i.e., assuming perfect sensitivity and specificity) expert dataset. In summary, our results highlight variation in the reliability of visual detection, and we provide an approach to calculate this and facilitate optimisation of risk-based surveillance strategies in plant health.

## Introduction

Biological invasions are a major cause of global ecosystem destabilisation, affecting both terrestrial and aquatic systems [1]. Such invasions are occurring with increasing frequency, and are projected to continue to increase towards 2050 [2], driven by factors such as human mediated transport, and those associated with environmental change (e.g., climate change and socio-economic activity) [3].

Invasive plant insect pests and pathogens (from hereon in collectively referred to as pests) are a pertinent example of the negative impacts of biological invasions. For example, *Xylella fastidiosa* was first detected in Europe in 2013, and identified as the causal agent of Olive Quick Decline Syndrome in Italy [4]. Genetic analyses revealed the bacterium was most likely introduced via ornamental coffee plants from Costa Rica [5], and is now projected to cost the industry in Italy alone up to ~ € 5 billion over a 50 year period [6].

As well as impacting agricultural systems, invasive plant pests can have a major impact on natural and semi-natural plant communities. *Hymenoscyphus fraxineus* is a fungus which is native to East Asia, and does not cause disease in its native range [7]. The fungus was introduced to Europe at least four decades ago [8], where it is the causal agent of ash dieback disease [9], which is leading to mortality of European native European ash (*Fraxinus excelsior*) and narrow-leaved ash (*Fraxinus angustifolia*) trees across the continent, with *F. excelsior* even being deemed at risk of extinction in northern Europe [10].

Once the epidemic spread of such invasive plant pests surpasses a certain threshold, eradication in the landscape may no longer be realistic [11]. Effective surveillance is essential for early detection, and this maximises the likelihood of eradication [12]. Even if eradication is not possible, early detection enables early intervention, which can reduce the overall costs and resources required for disease management [11]. Surveys for the detection of invading plant pests are often led by visual inspection, with confirmation via molecular diagnostics [13,14]. There is also an important role of citizen science in this process by increasing the number of potential reporters [15], which is highlighted by the discovery of *Dryocosmus kuriphilus* (oriental chestnut gall wasp) in the UK by a citizen scientist [16].

Although the importance of quantifying the reliability of laboratory diagnostic tests is recognised (e.g., through accreditation schemes [14] or requirements to validate regulatory diagnostics [17]), the probability of detecting a pest via initial visual inspection is rarely quantified. Without reliable knowledge of the sensitivity (probability of correctly identifying the presence of a pest) and specificity (probability of correctly identifying the absence of a pest) of visual inspection, it is impossible to answer key practical questions from survey data, such as: what prevalence has an invader reached when it is first detected [18] or what is the probability that a pest is absent if it is not detected by a survey [13].

The comparable lack of attention to the sensitivity and specificity of visual inspection is partly due to the rarity of 'gold-standard' reference datasets (i.e., assuming perfect sensitivity and specificity) to calculate these parameters in the field. In Australasia efforts to quantify the reliability of visual inspection have involved using man-made 'mimic' pests [19–22] and pest photo quizzes [23]. Recently, Vallee et al., [24] explored methods using Bayesian modelling to estimate the sensitivity and specificity of visual inspection and soil baiting in the absence of a 'gold-standard' dataset for surveys of *Phytophthora agathicida* on Kauri trees in the Waitākere Ranges in New Zealand, but this was not generalisable beyond the dataset in question.

Approaches to enable estimation of sensitivity and specificity in the absence of a 'gold-standard' that are applicable to wider plant health are sorely needed. In this study, we aimed to address this knowledge gap surrounding sensitivity and specificity for visual plant health surveys. To achieve this, we used acute oak decline (AOD) as a case study due to the availability of study sites containing trees with known status for three different signs and symptoms (from hereon in collectively referred to as symptoms). We conducted experiments using citizen scientists to assess and record symptoms of AOD across two oak woodlands in southern England. The availability of an expert who has monitored these study sites annually in detail for over a decade provided the closest available to a 'gold-standard' expert dataset for visual inspection of symptoms.

Next, we addressed the difficulty of calculating sensitivity and specificity more widely across different plant pests and diseases by examining an approach realistic for plant health to achieve this in the absence of a 'gold-standard' dataset. This involved first simulating survey datasets that were informed by empirical data from the AOD surveys to provide field-realistic estimates of parameter values for the sensitivity and specificity of visual inspection, as well as the number of trees that can be inspected by each of a given number of surveyors during an organised survey day. Datasets were simulated for 25 surveyors each inspecting either 80, 100 or 120 trees for two symptoms of disease at a site with a relatively higher true disease prevalence and at a site with a relatively lower true disease prevalence. Bayesian modelling was then applied to the simulated surveyor observation datasets for the two symptoms across the two sites to estimate the sensitivity and specificity of symptom detection (for the group of 25 surveyors and for individual surveyors) assuming prior knowledge of the sites with a relatively higher or lower true disease prevalence, but in the absence of 'gold-standard' expert data.

The specific objectives of this study were: (1) to provide an approach to quantify sensitivity and specificity in the absence of a 'gold-standard' dataset to facilitate optimisation of risk-based surveillance strategies in plant health; (2) to understand the extent to which sensitivity and specificity differ for visual inspection depending on the survey target (i.e., the symptom); (3) to understand to what extent surveyors differ in their sensitivity and specificity.

## Methods

### Ethics statement

Participants provided verbal consent to be part of the study, with all data analysed anonymously. The data are part of a multi-annual project assessing accuracy of volunteer reporting of AOD at the study sites, approved by the Forest Research Ethics Board.

### Acute oak decline as a case study

AOD is a decline disease involving patches of stem necrosis caused by a bacterial pathobiome and a native buprestid beetle that affects both UK native oak species, pedunculate oak (*Quercus robur*) and sessile oak (*Q. petraea*) [25–27]. The disease is characterised by the presence of four symptoms; externally visible weeping patches on oak stems, with underlying tissue necrosis in the sapwood, otherwise referred to as stem bleeds; the presence of cracks between bark plates with cavities of decayed tissue below; larval galleries of *Agrilus biguttatus* on more than 90% of affected trees, and externally visible D-shaped exit holes of the adult *A. biguttatus* on approximately a third of affected trees [25].

The presence of three externally visible characteristic symptoms (stem bleeds, bark cracks, *A. biguttatus* exit holes) (Fig 1), and the expert knowledge of AOD on the study sites enabled the design of surveys that could be used to calculate the sensitivity and specificity of citizen scientists at visually detecting morphologically different tree health symptoms.

### Acute oak decline surveys

Richmond Park (latitude: 51.4558, longitude: -0.2699) and Hatchlands Park (latitude: 51.2545, longitude: -0.4738) were selected for AOD surveys based on their good public accessibility for citizen scientist surveyors, and the presence of knowledge of tree health status from annual AOD monitoring by an expert for over a decade (since 2009 at Hatchlands Park and since 2010 at Richmond Park [28]). Walking routes of ~4 km were mapped on both sites, and oak trees along these routes were labelled and the expert assessed these trees for the presence of the three externally visible AOD symptoms one week prior to the first citizen scientist survey. In total, this involved 75 oak trees at Hatchlands Park and 100 at Richmond Park.

A total of 23 citizen scientists were recruited to conduct the survey, with nine individuals having worked as professional tree surveyors (i.e., experience working with trees, but not tree pest and disease professionals), and only three individuals

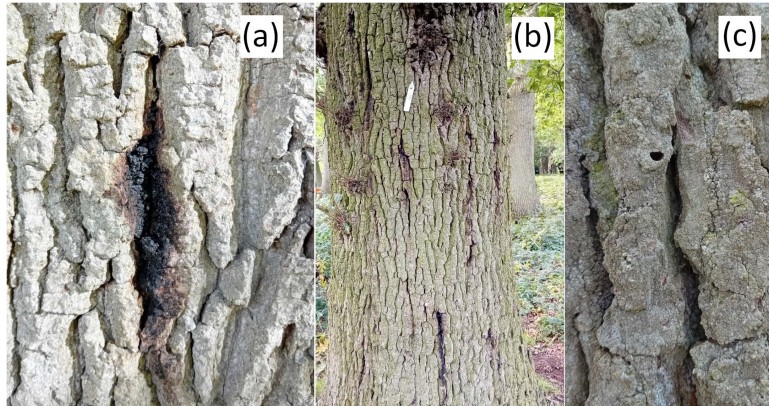

**Fig 1. Three characteristic externally visible symptoms of AOD [25].** (a) Weeping patch on oak trunk/ stem bleed; (b) AOD affected tree with stem bleeds and bark cracking between bark plates; (c) *Agrilus biguttatus* D-shaped exit hole (approximately 3-5 mm width). Photographs taken by M. Combes.

declaring no AOD knowledge (S1 File). All citizen scientist surveyors received detailed AOD training prior to and on the survey days (S1 File). On the survey days (Richmond Park: 20/07/2022, 01/08/2022; Hatchlands Park: 21/07/2022, 02/08/2022), they were instructed to individually walk the specified ~4 km routes and record the three external symptoms of AOD for each labelled tree (Fig 1). The expert assessor was present at each survey to note any changes in symptoms since their initial assessment.

## Analyses of acute oak decline survey data

The data on trees collected by citizen scientist surveyors was converted into reported presence and absence data for each of the three symptoms, and the sensitivity and specificity of symptom detection was then calculated for each symptom using the expert assessor as a reference, or 'gold-standard' (Equation 1). To do this, the number of true positives (*TPs*, where the expert and assessor both identified symptoms); false negatives (*FNs*, where the expert identified symptoms but the assessor did not); true negatives (*TNs*, where neither the expert nor the assessor identified symptoms); and false positives (*FPs*, where the expert did not identify symptoms but the assessor did) were identified and the equations below used to estimate sensitivity and specificity.

$$Sensitivity = \frac{TPs}{TPs + FNs} \tag{1}$$

$$Specificity = \frac{TNs}{TNs + FPs}$$

The relationship between sensitivity, the type of symptom, the individual surveyor and the frequency of the symptoms were first visualised and then statistically analysed using generalised linear models with a binomial distribution and logit link function. This was repeated to examine the relationship between specificity, the type of symptom and the individual surveyor. For both models, binary data was used for analyses, with the response variable corresponding to the probability of a true positive (*pTP*) for sensitivity models (Equation 2), and corresponding to the probability of a true negative (*pTN*) for specificity models (Equation 3).

$$
\begin{aligned}
logit\,(pTP) = {} & \beta_{se\_0} + \beta_{se\_1} Symptom\ type + \beta_{se\_2} frequency\ of\ a\ symptom\ on\ tree \\
& + \beta_{se\_3} Surveyor + \beta_{se\_4} Symptom\ type \bullet frequency\ of\ a\ symptom\ on\ a\ tree \\
& + \beta_{se\_5} Symptom\ type \bullet Surveyor
\end{aligned} \tag{2}
$$

$$logit(pTN) = \beta_{Sp\_0} + \beta_{sp\_1} Symptom\ type + \beta_{sp\_2} Surveyor + \beta_{sp\_3} Symptom\ type \bullet Surveyor \tag{3}$$

The significance of model parameters was assessed using type II likelihood ratio chi-square tests, with non-significant terms removed from models. Model interactions involving the individual surveyor and the frequency of the symptoms were not assessed due to a comparably limited dataset. Individual surveyors were included as a fixed effect in models rather than a random effect to statistically analyse the impact of the individual surveyors in our study on sensitivity and specificity, which is assumed to be a constant effect relating to a surveyor's ability. The data were analysed using R software version 4.1.2 [29], with packages car [30], DHARMa [31], DescTools [32], and visualised using ggplot2 [33].

## Methods to quantify sensitivity and specificity in the absence of a 'gold-standard' expert dataset

Using maximum likelihood [34] and Bayesian approaches [35], the sensitivity and specificity of diagnostic tests can be estimated when two (or more) tests are applied to the same individuals across two (or more) populations with different

true disease prevalences. These methods assume independence between the sensitivity of different diagnostic tests and independence between the specificity of different diagnostic tests (*cf.* Equation 6), although Dendukuri and Joseph [36] provide a methodology to account for non-independence (*cf.* Equation 5). Such non-independence arises when tests measure comparable biological processes of a disease that are therefore likely to have associated test errors [37] (see discussion for examples and S4 File for further detail). The application of both independence and non-independence models for veterinary disease diagnostics using a Bayesian approach outlined below, is described in detail by Branscum et al. [37]. We apply both these models to examine their utility for an approach realistic for visual plant health inspections, where in this context, each test refers to the visual assessment of a specific symptom by an individual surveyor, and we examine scenarios where two theoretical symptoms are assessed, rather than the three symptoms previously described.

As described in Branscum et al. [37], the data for tested hosts (trees in our case) after a survey, can be summarised in four categories: $y_{11}$ = test one is positive, test two is positive; $y_{12}$ = test one is positive, test two is negative; $y_{21}$ = test one is negative, test two is positive; $y_{22}$ = test one is negative, test two is negative. The number of observations in each of the four categories ($y_{11}$, $y_{12}$, $y_{21}$, $y_{22}$) following a survey of a given number of trees relates to the multinomial cell probabilities: $p_{11}$ = probability test one is positive, test two is positive; $p_{12}$ = probability test one is positive, test two is negative; $p_{21}$ = probability test one is negative, test two is positive; $p_{22}$ = probability test one is negative, test two is negative. For each population this can be characterised by Equation 4.

$$y\,(y_{11}, y_{12},\ y_{21},\ y_{22})\ \sim\ multinomial\,((p_{11},\ p_{12},\ p_{21},\ p_{22}),\ number\ of\ assessed\ trees) \tag{4}$$

The multinomial cell probabilities can be expressed using eight parameters if covariance parameters (to account for non-independence between both the sensitivity of different tests and between the and specificity of different tests; *cf.* S4 File), are included in the model (Equation 5), and six if these covariance parameters are not included in the model (Equation 6): True disease prevalence in population one, $\pi_1$; True disease prevalence in population two, $\pi_2$; Sensitivity of diagnostic test one, $Se_1$; Sensitivity of diagnostic test two, $Se_2$; Specificity of diagnostic test one, $Sp_1$; Specificity of diagnostic test two, $Sp_2$; Covariance between tests on disease positive hosts, $covD^+$; Covariance between tests for on disease negative hosts, $covD^-$.

$$p_{11} = \pi[Se_1 Se_2 + covD^+] + (1-\pi)[(1-Sp_1)(1-Sp_2) + covD^-], \tag{5}$$

$$p_{12} = \pi\left[Se_1(1-Se_2) - covD^+\right] + (1-\pi)[(1-Sp_1)Sp_2 - covD^-],$$

$$p_{21} = \pi\left[(1-Se_1)Se_2 - covD^+\right] + (1-\pi)[Sp_1(1-Sp_2) - covD^-],$$

$$p_{22} = \pi\left[(1-Se_1)(1-Se_2) + covD^+\right] + (1-\pi)[Sp_1 Sp_2 + covD^-].$$

$$p_{11} = \pi Se_1 Se_2 + (1-\pi)(1-Sp_1)(1-Sp_2), \tag{6}$$

$$p_{12} = \pi Se_1(1-Se_2) + (1-\pi)(1-Sp_1)Sp_2,$$

$$p_{21} = \pi(1-Se_1)Se_2 + (1-\pi)Sp_1(1-Sp_2),$$

$$p_{22} = \pi(1-Se_1)(1-Se_2) + (1-\pi)Sp_1 Sp_2.$$

Markov chain Monte Carlo (MCMC) methods were used to iterate across a range of values for the parameters to estimate the probability distribution of observing the recorded data for the given value of the parameter, whilst accounting for the

prior probability distribution of parameters [38]. This produces an estimate of the probability of a parameter value given the data (the posterior probability distribution) (Equation 7).

$$Probability(parameter\ value\ |\ data) \ \propto\ Probability(data\ |\ parameter\ value) \times Probability(parameter\ value) \qquad (7)$$

For this model, beta distributions were used to estimate the prior probability distributions for the true disease prevalence, and the test specificity and sensitivity [35,37]. The priors for covariance parameters were modelled using a generalised beta distribution with uniform prior probability distributions, based on the minimum and maximum possible values [36,37].

### An approach to estimate sensitivity and specificity of visual plant health inspection in the absence of 'gold-standard' data

Simulated survey datasets were used to explore field-realistic scenarios for an approach for the quantification of the sensitivity and specificity of plant health visual inspection in the absence of a 'gold-standard' expert dataset. The surveyor sensitivity and specificity values for the visual detection of two symptoms were generated from distributions based on the sensitivity and specificity for the visual detection of AOD stem bleed symptoms (Fig 2). To add non-independence between the detection of the two symptoms dependent on the disease status of a host (i.e., covariance between the sensitivity of tests and covariance between the specificity of tests), we assumed a probability of 0.85 that symptom two would be scored as positive if symptom one was scored as positive and the host was disease positive, and that symptom two would be scored as negative if symptom one was scored as negative and the host was disease negative. This value was chosen in the absence of prior information on the relationship between the detection probability for the visual inspection of different symptoms that can be expected in wider plant health, but was selected to test the robustness of using the covariance model, whilst also avoiding a more unrealistic scenario where the diagnostic result of two symptoms on a given plant are close to identical. The covariance values for simulated datasets (Equation 8), along with the minimum and maximum

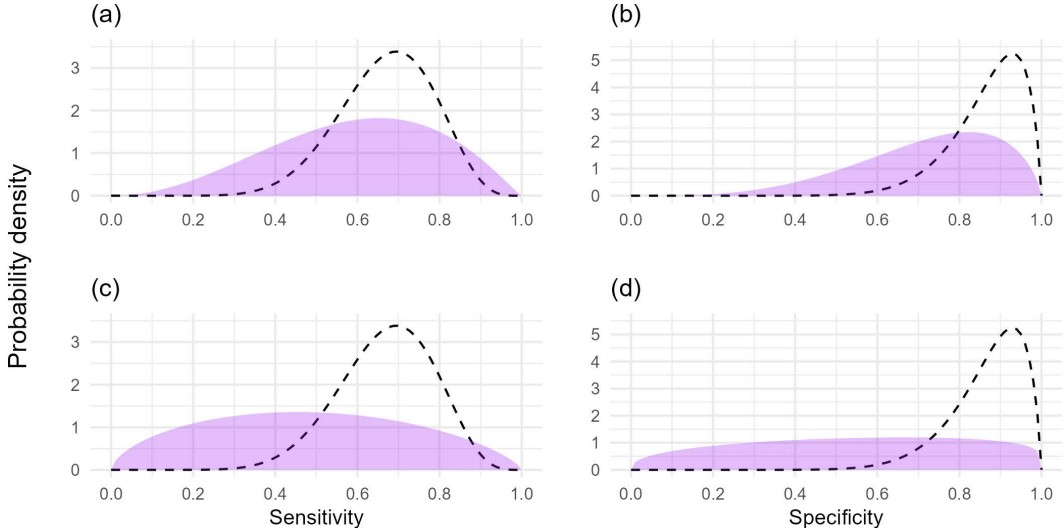

**Fig 2. Prior probability distributions of sensitivity and specificity and the distributions that the surveyor sensitivity and specificity values were generated from.** (a) Good prior knowledge of sensitivity, (b) good prior knowledge of specificity, (c) uninformed prior knowledge of sensitivity, (d) uninformed prior knowledge of specificity. The prior probability distributions are represented by the purple area, with the dashed lines representing the distribution used to generate the surveyor sensitivity and specificity values. The prior probability distributions for very good prior knowledge of sensitivity and specificity are not shown as separate figure panels because they are the same as the distributions that data were generated from (dashed lines).

possible values for the covariance parameters given the sensitivity and specificity values of simulated surveyors (Equation 9) are provided in S3 File. These values were calculated to provide the actual range of covariance values from simulated datasets, and the potential range of covariance values from simulated datasets given surveyor sensitivity and specificity (see S4 File for further details on derivation).

$$Se_{12} = Probability(Symptom\ 1\ positive,\ Symptom\ 2\ positive\ |\ Diseased\ host), \quad (8)$$

$$Sp_{12} = Probability(Symptom\ 1\ negative,\ Symptom\ 2\ negative\ |\ Disease\ free\ host),$$

$$covD^+ = Se_{12} - Se_1 Se_2,$$

$$covD^- = Sp_{12} - Sp_1 Sp_2.$$

$$Max\ covD^+ : \min(Se_1, Se_2) - Se_1 Se_2, \quad (9)$$

$$Min\ covD^+ : (Se_1 - 1)(1 - Se_2),$$

$$Max\ covD^- : \min(Sp_1, Sp_2) - Sp_1 Sp_2,$$

$$Min\ covD^- : (Sp_1 - 1)(1 - Sp_2).$$

Based on experience with the AOD surveys, it was believed ~25 surveyors can be organised to each inspect ~100 trees in a day. We therefore tested an approach with survey datasets simulated using the distributions of sensitivity and specificity for individuals detecting AOD stem bleeds (i.e., a symptom with moderate detection difficulty) either with or without covariance in the ability to detect each symptom (both based on sensitivity and specificity of AOD stem bleed detection) with the models above and 25 surveyors inspecting 80, 100 and 120 trees each from higher true disease prevalence, and lower true disease prevalence areas (see below for more detail on true prevalences). Surveyors then record the presence or absence of two disease symptoms on each assessed tree. This produces count data for $y_{11}$, $y_{12}$, $y_{21}$, $y_{22}$.

We anticipate informative prior knowledge of whether a location is the higher true disease prevalence or the lower true disease prevalence area (Fig 3 and S3 File). Informative prior knowledge of the true disease prevalence of these site locations can be reliably assumed by combining local expert knowledge of these sites with statistical methods that utilise knowledge of pest epidemiology. For example, a simple 'rule of thumb' approach can be used to estimate the true disease prevalence upon detection of a pest through knowledge of the epidemic growth rate when a fixed number of samples (or observations) are taken at known regular intervals [18].

Nevertheless, we explored scenarios that examined the impact of misspecified true disease prevalence prior distribution estimates on the of sensitivity and specificity estimates. The scenarios explored were: reliable knowledge of true disease prevalence, i.e., ~0.3 lower site (61.7 percentile on prior distribution) and ~0.6 higher site (48.3 percentile on prior distribution); underestimate of true disease prevalence, i.e., ~0.4 lower site (83.3 percentile on prior distribution) and ~0.8 higher site (91.4 percentile on prior distribution); overestimate of true disease prevalence, i.e., ~0.1 lower site (7.0 percentile on prior distribution) and ~0.4 higher site (10.0 percentile on prior distribution). Each of these true disease prevalence values varied by as much as ~0.1 from these values in simulated datasets (see S3 File for the actual true disease prevalence values from each simulation).

We assume no prior knowledge of the covariance parameters between symptoms, but we anticipate knowledge of whether covariance may be present. This means we only include covariance parameters when we simulate data with covariance. In the model, we specify uniform prior probability distributions between the minimum and maximum possible values (cf. [36,37]) (Eq 9 and S4 File).

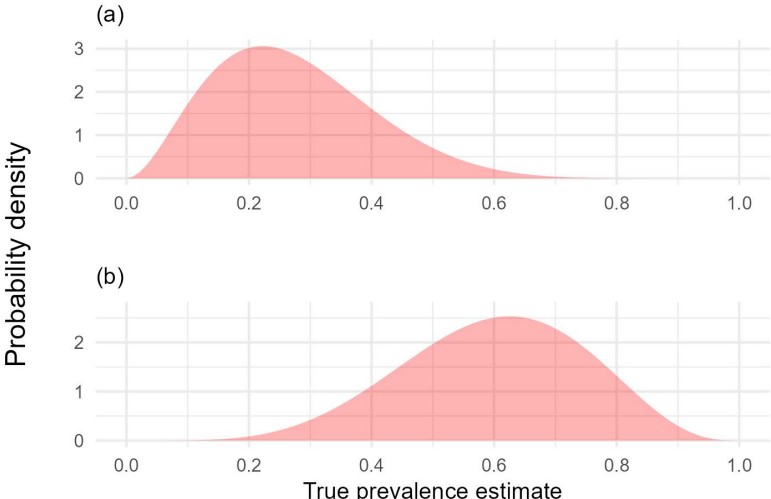

**Fig 3. Prior probability distributions of (a) lower, (b) higher true disease prevalence locations.** Surveyor observation datasets were simulated using a true disease prevalence of 0.3 for lower true disease prevalence locations and a true disease prevalence of 0.6 for higher true disease prevalence locations.

Three scenarios with different prior knowledge of the sensitivity and specificity for the two symptoms were tested, with the ability to estimate the sensitivity and specificity of symptom one assessed in subsequent analyses: (1) good prior knowledge of sensitivity and specificity of both symptoms; (2) poor prior knowledge of the sensitivity and specificity of symptom one, but very good prior knowledge of the sensitivity and specificity of symptom two; (3) poor prior knowledge of the sensitivity and specificity of both symptoms (Fig 2 and S3 File). These scenarios were examined (1) excluding covariance parameters with a dataset without simulated covariance, and (2) including covariance parameters on a dataset with simulated covariance. The impact of uncertainties of true disease prevalence estimates were then examined for models containing covariance parameters for each of the above three scenarios of prior knowledge of sensitivity and specificity.

For each surveyor, MCMC methods (using Gibbs sampling) were then used to iterate across a range of values for each model parameter. A single chain had 10,000 burn-in iterations, 2000 adaptive iterations and 500,000 sample iterations. To check model convergence, this was repeated with nine additional chains and the potential scale reduction factor of the Gelman-Rubin statistic autocorrelation of the sample (psrf) was calculated, with values above 1.1 indicating inadequate model convergence. Results were excluded from further analyses if model convergence was inadequate.

Posterior estimates of the alpha and beta parameters for the beta distribution of a surveyor's sensitivity and specificity were obtained for each iteration of the chain. These were then each thinned at every $10^{th}$ value after the initial 12,000 iterations, providing 48,800 alpha and beta value estimates for the beta distributions of sensitivity and specificity. The sensitivity and specificity of an individual surveyor was then estimated by randomly sampling values from each of these 48,800 possible beta distributions. The 48,800 estimates of sensitivity and specificity for each surveyor were combined to provide a group-level dataset of up to 1,220,000 values to estimate the group-level distribution of these parameters (i.e., distribution surveyors were sampled from). Kernel density estimates were then computed, and the results were visualised using a density plot.

Bayesian modelling and analyses were performed using Just another Gibbs Sampler (JAGS) program version 4.3.0 [39] in R software version 4.1.2 [29] with the packages runjags [40], coda [41], epiR [42] and fitdistrplus [43], and visualised using the ggplot2 [33].

## Results

### Acute oak decline case study

Calculation of surveyor sensitivity and specificity of visual detection of symptoms using expert data as a 'gold-standard' revealed large variation between the three AOD symptoms and between individual surveyors (Fig 4 and S2 File). Sensitivity differed between symptoms, and between surveyors, but increased with the frequency of a symptom on a tree (Fig 5). Significant interactions were present between both the type of symptom and frequency of a symptom on a tree (LR $\chi^2 = 49.803$, df = 2, p < 0.001), and between the type of symptom and the surveyor (LR $\chi^2 = 106.58$, df = 44, p < 0.001) (Equation 2). This highlights that the positive relationship between symptom frequency and sensitivity differs between symptoms, and that surveyor performance was not consistent relative to the symptom.

Specificity was higher than sensitivity for 22 of 23 surveyors, but also differed between symptoms, and surveyors, with a significant interaction present between the two (LR $\chi^2 = 192.19$, df = 44, p < 0.001) (Equation 3).

Overall, stem bleeds were the most reliable symptom to assess when accounting for both sensitivity and specificity, whilst *A. bigutattus* beetle exit holes had a high specificity but a low sensitivity (i.e., it was difficult to detect a D-shaped 3–5 mm exit hole on a tree and not common to falsely report one). Bark cracks were the least reliable symptom to visually detect (Fig 4), with many individuals performing similar to random chance (S1 Fig).

### An approach to estimate sensitivity and specificity in wider plant health

With only reliable prior knowledge of the true disease prevalence in 'higher' and 'lower' true disease prevalence locations the sensitivity and specificity of individual surveyors could be estimated with reasonable accuracy when as few as 80 trees were surveyed in each of the locations (Fig 6 and S3 File). A notable outlier is present in Fig 6b, and represents a

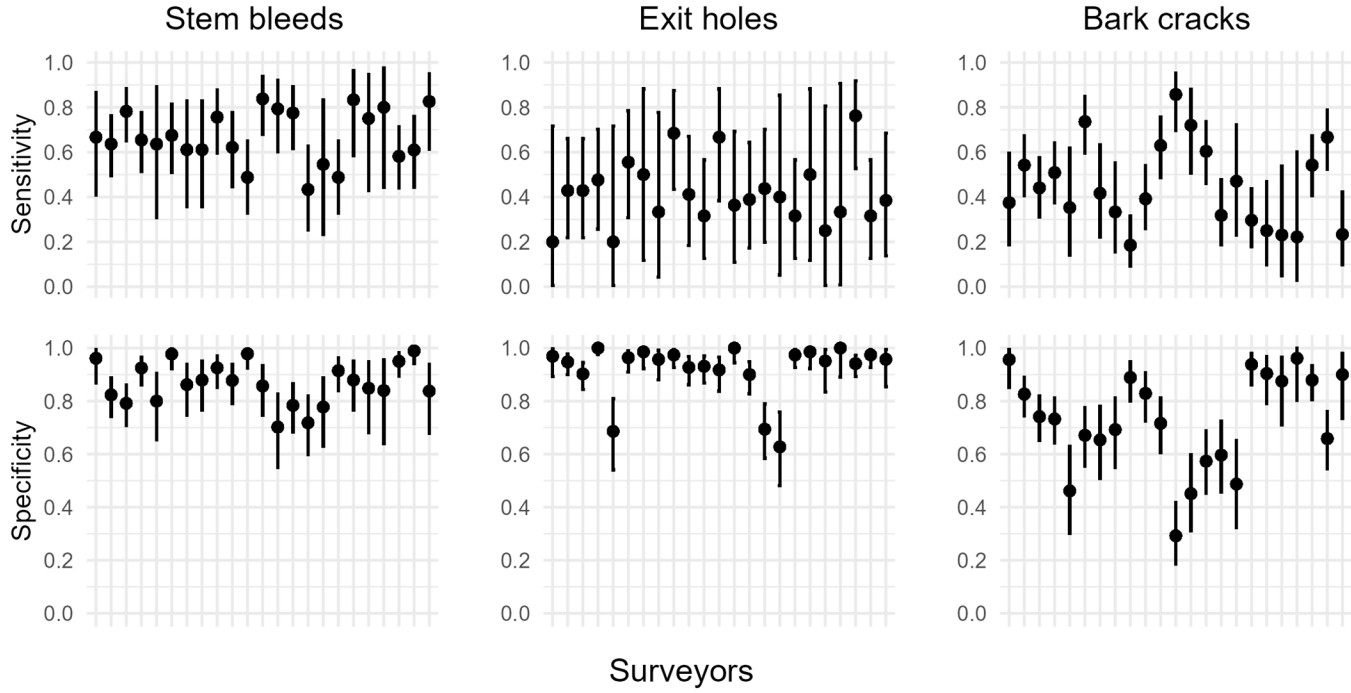

**Fig 4. Sensitivity and specificity of three externally visible AOD symptoms on the main stem.** The presence/ absence of bleeds/ weeping patches, *Agrilus* beetle exit holes and cracks between bark plates was assessed by 23 surveyors on up to 175 oak trees. Values of sensitivity and specificity were calculated using expert data as a 'gold-standard', with error bars representing 95% confidence intervals. Surveyors are ordered consistently between plots on the x axes.

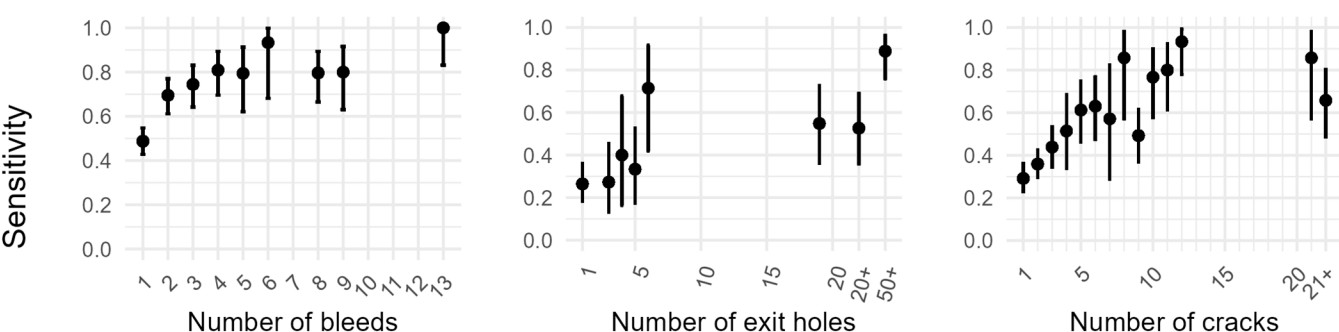

**Fig 5. Sensitivity of three externally visible AOD symptoms on the main stem against the frequency of each symptom on a host.** The presence/ absence of bleeds/ weeping patches, *Agrilus* beetle exit holes, cracks between bark plates was assessed by 23 surveyors on up to 175 oak trees. Data was pooled for surveyors, sensitivity was calculated using expert data as a 'gold-standard', and then compared against the number of symptoms scored on a tree from the expert data. Error bars represent 95% confidence intervals. Values on the x axes '20+', '50+', '21+' represent categorical data that were recorded at a lower precision due to the high frequency of symptoms on the host.

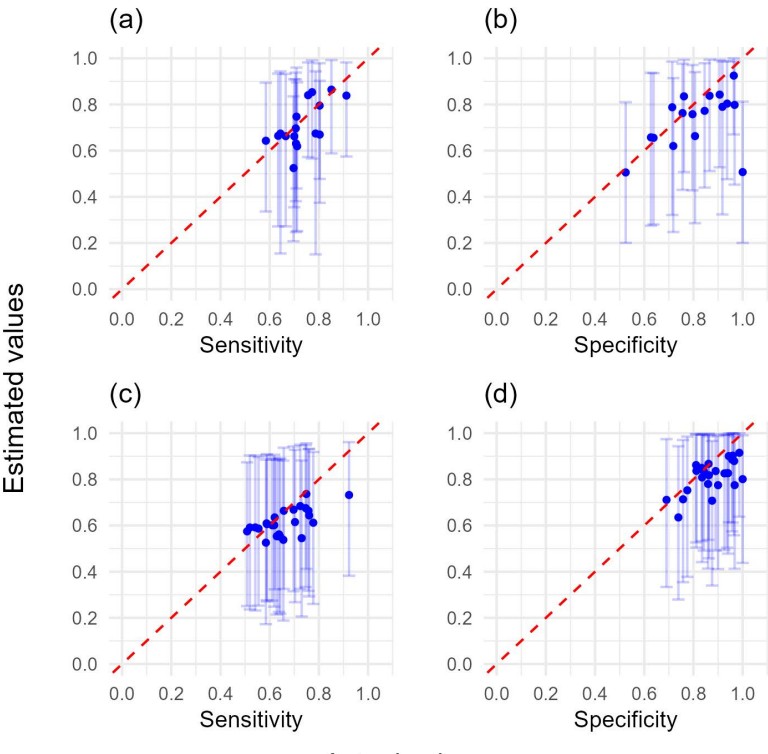

**Fig 6. The estimated sensitivity and specificity for individual simulated surveyors against their actual sensitivity and specificity values (80 trees, poor knowledge of both symptoms' sensitivity and specificity, reliable true disease prevalence knowledge).** Poor sensitivity and specificity prior distributions were used with reliable prior knowledge of true disease prevalence with (a, b) no covariance model with no simulated covariance, (c, d) covariance model with simulated covariance. In both instances 80 trees each were assessed in higher and lower true disease prevalence locations. Error bars represent 95% confidence intervals, and dashed line represents perfect agreement between estimated values and actual values.

simulation where true disease prevalence only differed by 0.1 between low and high true disease prevalence locations, and the surveyor specificity value was not supported by the prior probability distribution. Indeed, model estimates for individual surveyors were less reliable when surveyor sensitivity and specificity for either symptom was at a tail of a prior distribution (S18 Fig and S3 File), and occasionally resulted in models not converging (see below).

With reliable prior knowledge of the true disease prevalence in 'higher' and 'lower' true disease prevalence locations, the group-level distributions of sensitivity and specificity the individual surveyor values were generated from could be reliably estimated by combining sensitivity and specificity estimates from the 25 surveyors both in the presence and absence of covariance in the model and simulated data (Fig 7 and S3 File). However, unsatisfactory model convergence (psrf > 1.1) occurred for some models without covariance parameters with poor prior information of sensitivity and specificity (80 trees 17/ 25 converged; 100 trees 20/25 converged; 120 trees 18/25 converged) (S3 File). This reduced surveyor numbers used in analyses and affected the accuracy of distribution estimates (Fig 7a and 7b). Informative prior information of the sensitivity and specificity of one, or both symptoms improved the accuracy of model outputs, although narrower 'good' and 'very good' prior knowledge of sensitivity and specificity did not necessarily improve individual estimates of these parameters because some surveyors were at the tails of these distributions. There was also minimal additional benefit in surveying either 100 or 120 trees per higher or lower true prevalence location (S2–S17 Figs).

Our results demonstrate that misspecified true disease prevalence prior distributions are associated with greater error in the estimates of sensitivity and specificity. In the presence of poor prior information on the sensitivity and specificity of both symptoms, underestimating true disease prevalence (i.e., true disease prevalence values falling on the 83.3 and 91.4 percentiles of the prior distributions) resulted in slight underestimates of specificity, and slight overestimates of sensitivity for both individual surveyor and distribution values (Figs 8 and 9 and S3 File). As above, the 'good' and 'very good' prior knowledge of sensitivity and specificity did not necessarily increase the accuracy of individual surveyor estimates because

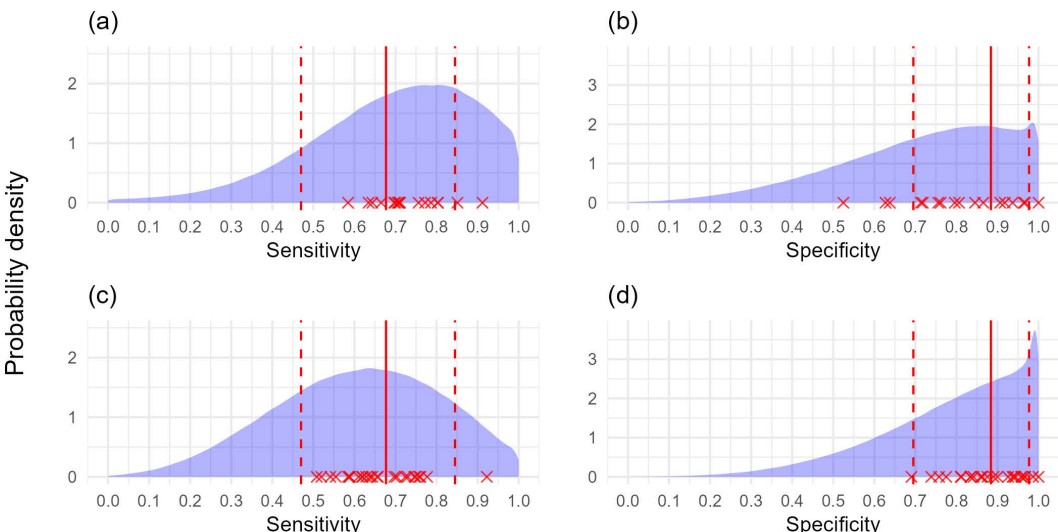

**Fig 7. Sensitivity and specificity distribution output using simulated survey datasets (80 trees, poor knowledge of both symptoms' sensitivity and specificity, reliable true disease prevalence knowledge).** Probability density of estimated sensitivity and specificity of surveyors using poor sensitivity and specificity prior distributions with reliable prior knowledge of true disease prevalence with (a, b) no covariance model with no simulated covariance, (c, d) covariance model with simulated covariance. In both instances 80 trees each were assessed in higher and lower true disease prevalence locations by 25 surveyors, but 8/25 models did not adequately converge for the no covariance model and were discarded. Red crosses represent true surveyor sensitivity and specificity. Solid red line represents the 50th percentile (median), dotted red lines represent the 5th and 95th percentiles of the distributions surveyor sensitivity and specificity values were generated from.

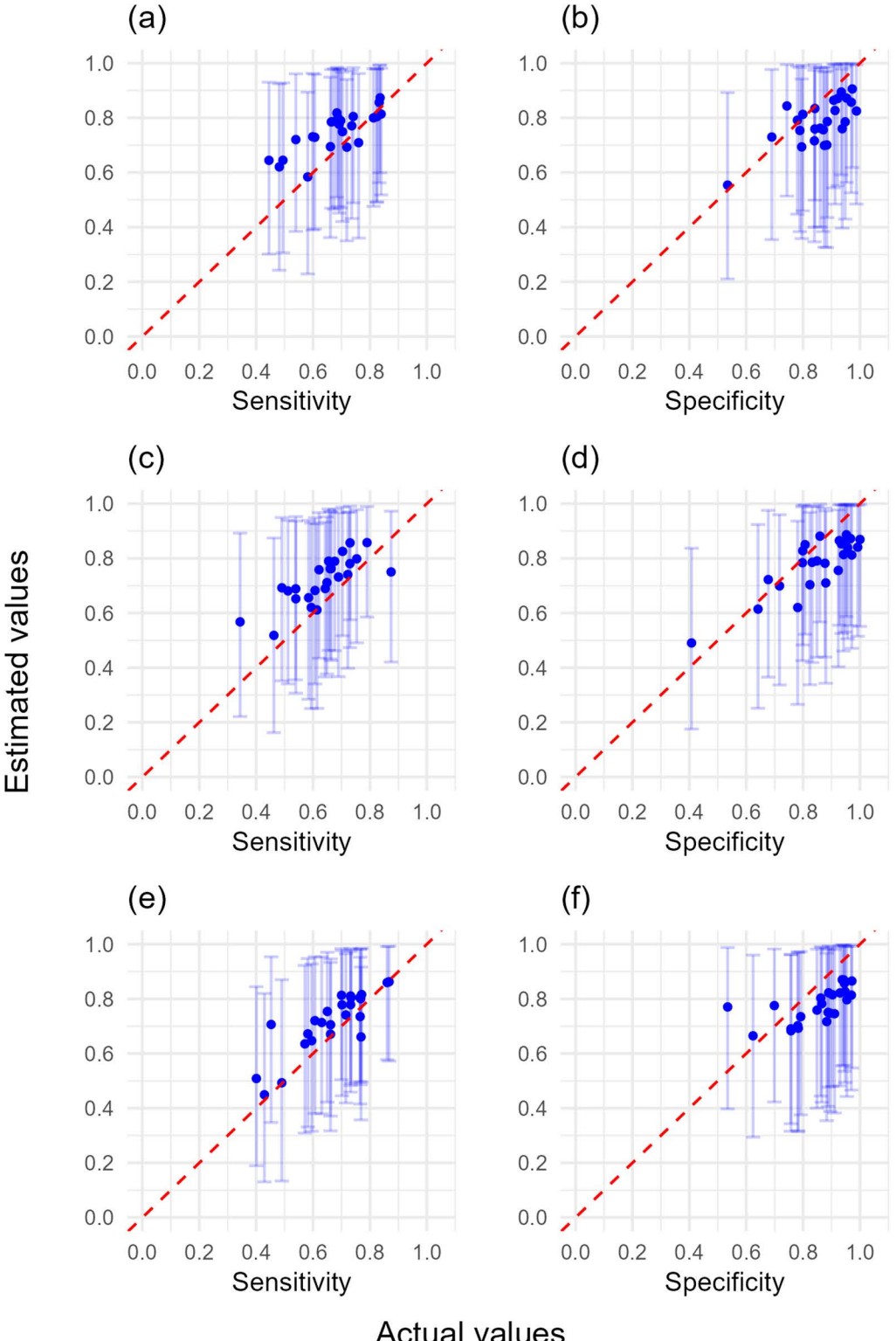

**Fig 8. The estimated sensitivity and specificity for individual simulated surveyors against their actual sensitivity and specificity values (poor knowledge of both symptoms' sensitivity and specificity, true disease prevalence underestimated).** Poor sensitivity and specificity prior distributions were used with misspecified prior knowledge which underestimated the true disease prevalence at survey locations, with surveyors assessing (a,

b) 80, (c, d) 100, or (e, f) 120 trees in each of the higher and lower true disease prevalence locations. Error bars represent 95% confidence intervals, and dashed line represents perfect agreement between estimated values and actual values.

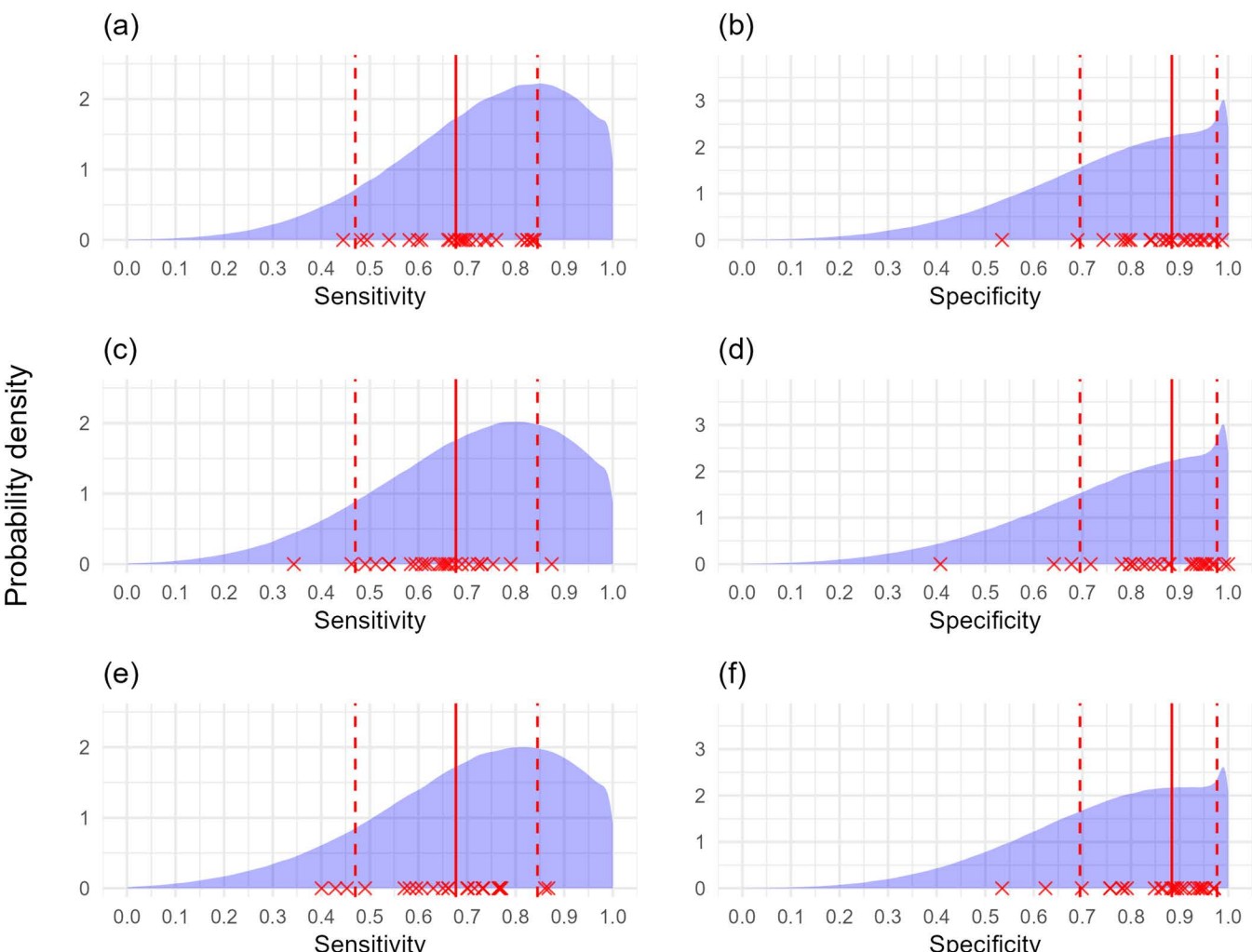

**Fig 9. Sensitivity and specificity distribution output using simulated survey datasets (poor knowledge of both symptoms' sensitivity and specificity, true disease prevalence underestimated).** Probability density of estimated sensitivity and specificity of surveyors using poor sensitivity and specificity prior distributions with misspecified prior knowledge which underestimated the true disease prevalence at survey locations, with 25 surveyors each assessing (a, b) 80, (c, d) 100, or (e, f) 120 trees in each of the higher and lower true disease prevalence locations. Red crosses represent true surveyor sensitivity and specificity. Solid red line represents the 50th percentile (median), dotted red lines represent the 5th and 95th percentiles of the distributions surveyor sensitivity and specificity values were generated from.

some surveyors were at the tails of these narrower distributions, and there was minimal benefit in surveying either 80, 100, or 120 trees per site (Figs 8, 9 and S19–S22 Figs and S3 File).

In contrast, overestimating true disease prevalence (i.e., true disease prevalence values falling on the 7.0 and 10.0 percentiles of the prior distributions) resulted in larger error in estimates of individual surveyor sensitivity, with underestimates of the distribution values of surveyor sensitivity (S23–S28 Figs and S3 File). Greater prior knowledge of surveyor

sensitivity and specificity increased the accuracy of sensitivity distribution estimates, but had a negligible effect on the accuracy of individual surveyor sensitivity estimates. Surveying either 80, 100 or 120 trees had negligible effect on estimates of both individual and distribution values. However, surveyor specificity could be reliably estimated regardless of prior information or whether 80, 100, or 120 trees were surveyed (S23–S28 Figs and S3 File). These results likely reflect the presence of insufficient disease positive observation data for reliable inference of sensitivity values due to the very low true disease prevalence on the 'lower site' (~0.1).

## Discussion

This study provides an approach to derive quantitative estimates of the sensitivity and specificity of visual inspection for symptoms in plant health in the absence of a 'gold-standard' reference dataset. This meets a key knowledge gap, since visual inspection is a cornerstone of plant health surveys, and lack of estimates for these parameters severely hinders adequate survey planning, interpretation of surveillance data, and instigation of effective outbreak management. Our findings highlight variation between individual surveyors and between different symptoms, thus emphasising the necessity of quantifying these parameters to optimise surveillance strategies in plant health. To facilitate this, our results demonstrate the utility of an approach utilising Bayesian modelling to provide estimates of the sensitivity and specificity in the absence of 'gold-standard' data and lack of informative prior information of these parameters.

Our results suggest fundamental surveillance metrics such as the number of visual assessments required to confidently declare an area as pest free, understanding of the true pest prevalence given a positive detection, and reliability of positive detection [12] will vary depending on the visual symptom of the pest. Previous work highlights that the sensitivity of a detection method affects the optimal deployment of surveillance strategies for early detection of a pest or disease in the landscape, with lower sensitivity leading to surveillance focussed in higher-risk locations [44]. The variation in sensitivity between different symptoms in our results highlights that in plant health, not only the type of detection method (e.g., insect trapping vs visual detection), but the particular visual symptom on a host will influence the optimal survey locations for early detection. For example, early detection surveillance for symptoms that are associated with a lower sensitivity of detection (e.g., exit holes) are more likely to be most effective by focussing surveys at higher-risk locations (rather than being spread more widely across the landscape) compared to symptoms that are associated with a relatively higher sensitivity of detection (e.g., stem bleeds). Our findings agree with previous work from Australasia which have found large variations in the reliability of visual detection depending on the survey target [19,20,23,45], with more visually distinct [20,45], or more familiar survey targets more likely to be reported [23]. Our results build on this by providing sensitivity and specificity estimates for different common tree health symptoms from field surveys using long-term expert data as a 'gold-standard', rather than using mimic objects [19,20], online quizzes [23], or passive surveillance reporting data [45].

In addition to informing AOD surveillance, in combination with epidemic spread models, the estimates of sensitivity presented in this study can be used to inform surveillance strategies for other plant pests and diseases of major concern. For example, the emerald ash borer (EAB) beetle is an *Agrilus* species that has killed millions of ash trees in the US, and also presents with D-shaped exit holes [46] that are directly comparable to *A. biguttatus* exit holes associated with AOD. *Phytophthora* spp., are a major cause of forest disease globally, and frequently present with bleeding canker symptoms directly comparable to those examined in this study [47]. However, the sensitivity presented for the AOD symptoms represents the sensitivity of detecting symptoms. Therefore, for application to other pests and diseases both the asymptomatic period and frequency of asymptomatic hosts must be considered.

Our results also demonstrate that sensitivity of symptom detection is greater when hosts are more affected by a pest (i.e., greater frequency of a symptom on a tree). First, this means that the difficulty of early detection of pests in the landscape due to low abundance will be compounded by a lower sensitivity of detection of initial symptoms of a pest. Second, these results demonstrate that the pathology of a disease must be considered to optimally apply results.

Importantly, the surveyors in our study were citizen scientists, and not tree pest and disease professionals. Citizen scientists involved in the study received the same training prior to conducting AOD surveys, but still varied greatly in their ability to visually detect symptoms. Citizen science has an important role in plant health [15], yet practically applicable information about how surveyors with different experience differ in their ability to detect plant pests and diseases is scarce. Previous survey data using mimic pests highlights that individuals classified as inexperienced were ~0.1 less likely to find a mimic pest, whilst the probability of false positives for individuals classified as inexperienced was much higher for some mimic pest patterns than others (~0.2 vs negligible difference) [19]. More recent work on visual inspection of Phony Peach Disease using qPCR testing as a 'gold-standard' dataset, found a difference in misclassification probability of ~0.1 between surveyors familiar with symptoms and those not familiar with symptoms, which was driven by lower false negative rates in the more experienced group [48].

Pocock et al. [49] found that citizen scientist data agreed well with expert data when recording *Cameraria ohridella* leaf damage scores and counts of *C. ohridella* adult moths, but not when counting the rarer associated parasitoid wasp, which was attributed to a lack of familiarity with the rarer species. Moreover, previous work investigating the collection of phenological data, found that expert surveyors had less variability compared to citizen scientists, but citizen scientists (regardless of training) performed similar to the expert surveyors when recording the presence of abundant flowers and fruits [50].

Together, in the context of plant health, these results suggest that frequent training would be beneficial for inexperienced surveyors, or for symptoms that are less commonly encountered. It has been shown that landowner reporting data and professional survey data provide similar estimates of AOD distribution data in the UK, and professional monitoring can be more resource efficient by using citizen science reports to guide surveys [51]. However, better understanding of the variation in surveyors' ability (which is also strongly influenced by time per inspection [22]), and more specific applicable information on the effect of training and experience on the detection of symptoms are required for optimised survey design, and effective integration of different types of citizen science data with professional surveys.

The difficulty obtaining 'gold-standard' survey datasets prohibits the routine quantification of the sensitivity and specificity of visual inspection in plant health. Estimates of these parameters are possible through studies using mimic pests and symptoms [19,20,22] or inoculation studies. However, inoculation studies are practically difficult to undertake for long-lived hosts (e.g., trees) and diseases that have multi-factorial causes or develop over a prolonged period (e.g., AOD). It is also challenging to ensure that the disease development from both controlled inoculations and mimic pests and symptoms are representative of disease development in the field, and that the presence of other ailments that can be mistaken for a pest or disease are appropriately represented under more controlled experimental conditions.

In our study, the expert survey data was assumed to be 'gold-standard' because this expert has more than a decade of experience surveying these trees for AOD [28]. It is possible that the expert did not classify all trees correctly, although the high performance of some citizen scientists with comparatively limited experience (Fig 4), suggests the expert will have sensitivity and specificity of at least >0.90, which should have minimal impact on the findings. However, the sensitivity and specificity values from the AOD survey represent the probability of correctly classifying symptom presence or absence, rather than pest or disease presence or absence because the sensitivity and specificity parameters were calculated relative to expert visual detection of the symptoms.

The approach outlined in this paper provides a solution to the problem of obtaining a 'gold-standard' survey dataset. Previous work used Bayesian modelling to estimate the sensitivity and specificity of both visual detection and soil baiting detection of *Phytophthora agathidicida* from Kauri trees in the Waitākere Ranges in New Zealand [24]. This was achieved by surveying 189 trees in high true disease prevalence areas and 572 trees in low true disease prevalence areas, and assuming independence of diagnostic tests (Equation 6) and prior knowledge of the sensitivity and specificity of soil baiting [24]. We build on this work by detailing an approach that is applicable to wider plant health by only using visual inspection from a fewer number of assessed trees, allowing non-independence between the assessed symptoms with poor prior

knowledge of sensitivity and specificity, and validating our results with simulated survey datasets that can be applied to different survey scenarios.

We show that with 25 surveyors, assessing a minimum of 80 plants each for two symptoms in relatively lower and relatively higher true disease prevalence locations, we can estimate the distribution of sensitivity and specificity for symptoms. Reliable prior knowledge of true disease prevalence in relatively higher and lower true disease prevalence survey areas provides sufficient information for Bayesian inference when combined with the observation data for two symptoms on the same plants, across the two survey areas and expressed as Equations 5 or Equation 6. Knowledge of the true disease prevalence of sites can be reliably assumed through a combination of local expert knowledge, and the use of statistical approaches that utilise knowledge of pest epidemiology, such as the simple 'rule of thumb approach' that only requires knowledge of the epidemic growth rate when a fixed number of samples (or observations) are taken at known regular intervals [18]. It is also possible to estimate the sensitivity and specificity of individual surveyors, but it is not yet known how individual surveyors vary between surveys, and it would be practically and financially challenging to quantify each individual surveyor that will be visually inspecting plants.

The approach outlined in this study can be applied by plant health authorities through a one-off organised survey event within the current range of a pest in higher and lower true prevalence sites. Extremes of true disease prevalence (i.e., <0.1 or >0.9) should be avoided to enable adequate observation data for reliable inference. The survey event will involve at least 25 surveyors that will record two prespecified symptoms of the pest of interest on each plant (at least 80 plants in each site). The minimum prior information required for subsequent modelling is knowledge of higher, or lower true disease prevalence sites, although greater prior knowledge of other parameters improves model accuracy. The output will provide an estimate of the distribution of sensitivity and specificity for this group of surveyors, to then be applied to different groups of surveyors with a similar level of experience. It is recommended surveyors have similar experience given that distributions of sensitivity and specificity will likely vary with experience. These estimates of parameter distributions can then be used to determine fundamental surveillance metrics [12] described earlier, as well as for optimisation of survey locations to maximise the likelihood of early detection [44].

The sensitivity output represents the probability of correctly identifying pest or disease presence from visual inspection for a given symptom within the surveyed areas. Post-hoc analyses utilising knowledge of the pathology and epidemiology of a disease can therefore be used to increase the accuracy of sensitivity estimates for application to other areas by accounting for spatial and temporal variation in both the expected proportions of asymptomatic hosts and symptom severity. Unreliable model estimates were obtained in our analyses when sensitivity and specificity values were at the tail of prior distributions, but this can be addressed by appropriate specification of prior information in Bayesian analyses using a combination of expert elicitation approaches [52], 'common biological sense' and sensitivity analyses of priors [53]. Misspecified prior knowledge of true disease prevalence on sites also resulted in less reliable estimates of sensitivity and specificity. Nonetheless, our results support that sensitivity and specificity estimates are reasonably robust to variation in true disease prevalence values provided that the prior distributions for the true disease prevalences are not misspecified (i.e., resulting in values at the tails of distributions). The misspecification of true disease prevalence prior distributions can be avoided by combining expert knowledge of sites with statistical methods to estimate true disease prevalence [18], in addition to using the appropriate methods to specify prior distributions [53].

Previous work demonstrates that models which don't account for non-independence of sensitivity and specificity between tests by failing to include covariance parameters in the model perform poorly [37]. This non-independence is likely to be present when tests examine comparable biological aspects of a disease [37]. For example, the sensitivity and specificity of detection of a pest from a necrotic lesion via an ELISA test for a pest antigen and via qPCR for pest DNA are unlikely to be independent, whereas examining the presence of soil samples for the presence of a pest (e.g., soil sampling for *Phytophthora* spp.), and visual inspection for a *Phytophthora* spp. induced stem bleed on a host are likely to be independent. An example of independence between visual symptoms on a tree could be the detection of canopy dieback,

and the detection of stem bleeds on a tree. Both these symptoms are associated with different biological mechanisms that could lead to false positive detection and false negative detection (see S4 File for further information).

However, our results highlight that including additional covariance parameters in the model incurs negligible penalty in the example used here. This likely reflects the benefit of informative prior knowledge of the true disease prevalence on sites, the fact that the true disease prevalence was very different between sites, and that a large number of hosts were surveyed on each site (at least 80 trees). Given the approach presented in this paper can provide sensitivity and specificity estimates with the presence of covariance in the data and model, and a potentially large penalty is incurred in models which falsely exclude covariance [37], we recommend including a covariance parameter in the model unless convincing data suggests otherwise. In future, more informative prior information on the covariance parameters for specific systems would strengthen model performance.

## Conclusions

This study demonstrates that the sensitivity and specificity of visual inspection varies between plant health symptoms and between surveyors. The data generated in our study for the inspection of AOD symptoms can be applied to surveillance for pests which present similar symptoms, such as EAB and *Phytophthora* spp. However, given the large variation present between surveyors, and likely impact of training and experience, results should only be applied to surveyors with similar experience to those in our study (i.e., trained citizen scientists), and using distributions of sensitivity and specificity that incorporate uncertainty estimates of these parameters. Future work should investigate this large surveyor variation, and understand the effect of training and experience of surveyors to allow better integration of citizen science activities with professional surveys. Moreover, given the ability to detect symptoms improves with symptom frequency, further work investigating the temporal pattern of symptom appearance (including asymptomatic period) would be highly valuable for plant health surveillance. The variation in sensitivity and specificity between symptoms and surveyors in this study means that: (1) the reliability of survey reports will vary between pests and surveyors; (2) different numbers of survey records will be required to declare different pests absent with the same confidence; (3) the true incidence given a positive detection will depend on the surveyor and symptom of the pest, not just the epidemiology; (4) survey locations to maximise the likelihood of early detection will differ depending on the visual symptom of a pest. To apply this knowledge, the approach presented in this paper enables quantification of the sensitivity and specificity for visual plant health surveys in the absence of rarely available 'gold-standard' datasets. This addresses a major knowledge gap, which will enable the calculation of fundamental surveillance metrics, and improve risk-based surveillance strategies in plant health. Further adaptation of these methods can be used for a broad range of visual environmental surveillance activities, as well as the ground truthing of novel surveillance technologies (such as remote sensing [54]) that rely on observation data.

## Supporting information

**S1 File. Details on pre-survey AOD training and citizen scientists' tree and tree health experience scored from a questionnaire.**
(PDF)

**S2 File. Citizen scientist surveyor sensitivity and specificity values.**
(XLSX)

**S3 File. Output statistics from model runs (individuals and distributions) with mean squared error values, and details of the prior probability distributions.**
(XLSX)

**S4 File. Further explanation of covariance parameters used in models.**
(PDF)

**S1 Fig. Receiver operating characteristic plots for three externally visible AOD symptoms.** The presence/ absence of bleeds/ weeping patches on the main stem, *Agrilus* beetle exit holes and cracks between bark plates were assessed by 23 surveyors on up to 175 oak trees. True positive rate and false positive rate were calculated using expert data as a 'gold-standard'. Error bars represent 95% confidence intervals.
(TIF)

**S2 Fig. Sensitivity and specificity distribution output using simulated survey datasets (80 trees, very good knowledge of one symptom's sensitivity and specificity, reliable true disease prevalence knowledge).** Probability density of estimated sensitivity and specificity of surveyors using poor sensitivity and specificity prior distributions for symptom one, and very good sensitivity and specificity prior distributions for symptom two with reliable prior knowledge of true disease prevalence with (a, b) no covariance model with no simulated covariance, (c, d) covariance model with simulated covariance. In both instances 80 trees each were assessed in higher and lower true disease prevalence locations by 25 surveyors. Red crosses represent true surveyor sensitivity and specificity. Solid red line represents the 50th percentile (median), dotted red lines represent the 5th and 95th percentiles of the distributions surveyor sensitivity and specificity values were generated from.
(TIF)

**S3 Fig. The estimated sensitivity and specificity for individual simulated surveyors against their actual sensitivity and specificity values (80 trees, very good knowledge of one symptom's sensitivity and specificity, reliable true disease prevalence knowledge).** Poor sensitivity and specificity prior distributions for symptom one, and very good sensitivity and specificity prior distributions for symptom two were used with reliable prior knowledge of true disease prevalence with (a, b) no covariance model with no simulated covariance, (c, d) covariance model with simulated covariance. In both instances 80 trees each were assessed in higher and lower true disease prevalence locations. Error bars represent 95% confidence intervals, and dashed line represents perfect agreement between estimated values and actual values.
(TIF)

**S4 Fig. Sensitivity and specificity distribution output using simulated survey datasets (80 trees, good knowledge of both symptoms' sensitivity and specificity, reliable true disease prevalence knowledge).** Probability density of estimated sensitivity and specificity of surveyors using good sensitivity and specificity prior distributions with reliable prior knowledge of true disease prevalence with (a, b) no covariance model with no simulated covariance, (c, d) covariance model with simulated covariance. In both instances 80 trees each were assessed in higher and lower true disease prevalence locations by 25 surveyors. Red crosses represent true surveyor sensitivity and specificity. Solid red line represents the 50th percentile (median), dotted red lines represent the 5th and 95th percentiles of the distributions surveyor sensitivity and specificity values were generated from.
(TIF)

**S5 Fig. The estimated sensitivity and specificity for individual simulated surveyors against their actual sensitivity and specificity values (80 trees, good knowledge of both symptoms' sensitivity and specificity, reliable true disease prevalence knowledge).** Good sensitivity and specificity prior distributions were used with reliable prior knowledge of true disease prevalence with (a, b) no covariance model with no simulated covariance, (c, d) covariance model with simulated covariance. In both instances 80 trees each were assessed in higher and lower true disease prevalence locations. Error bars represent 95% confidence intervals, and dashed line represents perfect agreement between estimated values and actual values.
(TIF)

**S6 Fig. Sensitivity and specificity distribution output using simulated survey datasets (100 trees, poor knowledge of both symptoms' sensitivity and specificity, reliable true disease prevalence knowledge).** Probability density of estimated sensitivity and specificity of surveyors using poor sensitivity and specificity prior distributions with reliable prior knowledge of true disease prevalence with (a, b) no covariance model with no simulated covariance, (c, d) covariance model with simulated covariance. In both instances 100 trees each were assessed in higher and lower true disease prevalence locations by 25 surveyors, but 5/25 models did not adequately converge for the no covariance model and were discarded. Red crosses represent true surveyor sensitivity and specificity. Solid red line represents the 50th percentile (median), dotted red lines represent the 5th and 95th percentiles of the distributions surveyor sensitivity and specificity values were generated from.
(TIF)

**S7 Fig. The estimated sensitivity and specificity for individual simulated surveyors against their actual sensitivity and specificity values (100 trees, poor knowledge of both symptoms' sensitivity and specificity, reliable true disease prevalence knowledge).** Poor sensitivity and specificity prior distributions were used with reliable prior knowledge of true disease prevalence with (a, b) no covariance model with no simulated covariance, (c, d) covariance model with simulated covariance. In both instances 100 trees each were assessed in higher and lower true disease prevalence locations. Error bars represent 95% confidence intervals, and dashed line represents perfect agreement between estimated values and actual values.
(TIF)

**S8 Fig. Sensitivity and specificity distribution output using simulated survey datasets (100 trees, very good knowledge of one symptom's sensitivity and specificity, reliable true disease prevalence knowledge).** Probability density of estimated sensitivity and specificity of surveyors using poor sensitivity and specificity prior distributions for symptom one, and very good sensitivity and specificity prior distributions for symptom two with reliable prior knowledge of true disease prevalence with (a, b) no covariance model with no simulated covariance, (c, d) covariance model with simulated covariance. In both instances 100 trees each were assessed in higher and lower true disease prevalence locations by 25 surveyors. Red crosses represent true surveyor sensitivity and specificity. Solid red line represents the 50th percentile (median), dotted red lines represent the 5th and 95th percentiles of the distributions surveyor sensitivity and specificity values were generated from.
(TIF)

**S9 Fig. The estimated sensitivity and specificity for individual simulated surveyors against their actual sensitivity and specificity values (100 trees, very good knowledge of one symptom's sensitivity and specificity, reliable true disease prevalence knowledge).** Poor sensitivity and specificity prior distributions for symptom one, and very good sensitivity and specificity prior distributions for symptom two were used with reliable prior knowledge of true disease prevalence with (a, b) no covariance model with no simulated covariance, (c, d) covariance model with simulated covariance. In both instances 100 trees each were assessed in higher and lower true disease prevalence locations. Error bars represent 95% confidence intervals, and dashed line represents perfect agreement between estimated values and actual values.
(TIF)

**S10 Fig. Sensitivity and specificity distribution output using simulated survey datasets (100 trees, good knowledge of both symptoms' sensitivity and specificity, reliable true disease prevalence knowledge).** Probability density of estimated sensitivity and specificity of surveyors using good sensitivity and specificity prior distributions with reliable prior knowledge of true disease prevalence with (a, b) no covariance model with no simulated covariance, (c, d) covariance model with simulated covariance. In both instances 100 trees each were assessed in higher and lower true disease prevalence locations by 25 surveyors. Red crosses represent true surveyor sensitivity and specificity. Solid red line

represents the 50th percentile (median), dotted red lines represent the 5th and 95th percentiles of the distributions surveyor sensitivity and specificity values were generated from.
(TIF)

**S11 Fig. The estimated sensitivity and specificity for individual simulated surveyors against their actual sensitivity and specificity values (100 trees, good knowledge of both symptoms' sensitivity and specificity, reliable true disease prevalence knowledge).** Good sensitivity and specificity prior distributions were used with reliable prior knowledge of true disease prevalence with (a, b) no covariance model with no simulated covariance, (c, d) covariance model with simulated covariance. In both instances 100 trees each were assessed in higher and lower true disease prevalence locations. Error bars represent 95% confidence intervals, and dashed line represents perfect agreement between estimated values and actual values.
(TIF)

**S12 Fig. Sensitivity and specificity distribution output using simulated survey datasets (120 trees, poor knowledge of both symptoms' sensitivity and specificity, reliable true disease prevalence knowledge).** Probability density of estimated sensitivity and specificity of surveyors using poor sensitivity and specificity prior distributions with reliable prior knowledge of true disease prevalence with (a, b) no covariance model with no simulated covariance, (c, d) covariance model with simulated covariance. In both instances 120 trees each were assessed in higher and lower true disease prevalence locations by 25 surveyors, but 7/25 models did not adequately converge for the no covariance model and were discarded. Red crosses represent true surveyor sensitivity and specificity. Solid red line represents the 50th percentile (median), dotted red lines represent the 5th and 95th percentiles of the distributions surveyor sensitivity and specificity values were generated from.
(TIF)

**S13 Fig. The estimated sensitivity and specificity for individual simulated surveyors against their actual sensitivity and specificity values (120 trees, poor knowledge of both symptoms' sensitivity and specificity, reliable true disease prevalence knowledge).** Poor sensitivity and specificity prior distributions with reliable prior knowledge of true disease prevalence with (a, b) no covariance model with no simulated covariance, (c, d) covariance model with simulated covariance. In both instances 120 trees each were assessed in higher and lower true disease prevalence locations. Error bars represent 95% confidence intervals, and dashed line represents perfect agreement between estimated values and actual values.
(TIF)

**S14 Fig. Sensitivity and specificity distribution output using simulated survey datasets (120 trees, very good knowledge of one symptom's sensitivity and specificity, reliable true disease prevalence knowledge).** Probability density of estimated sensitivity and specificity of surveyors using poor sensitivity and specificity prior distributions for symptom one, and very good sensitivity and specificity prior distributions for symptom two with reliable prior knowledge of true disease prevalence with (a, b) no covariance model with no simulated covariance, (c, d) covariance model with simulated covariance. In both instances 120 trees each were assessed in higher and lower true disease prevalence locations by 25 surveyors. Red crosses represent true surveyor sensitivity and specificity. Solid red line represents the 50th percentile (median), dotted red lines represent the 5th and 95th percentiles of the distributions surveyor sensitivity and specificity values were generated from.
(TIF)

**S15 Fig. The estimated sensitivity and specificity for individual simulated surveyors against their actual sensitivity and specificity values (120 trees, very good knowledge of one symptom's sensitivity and specificity, reliable true disease prevalence knowledge).** Poor sensitivity and specificity prior distributions for symptom one, and very

good sensitivity and specificity prior distributions for symptom two were used with reliable prior knowledge of true disease prevalence with (a, b) no covariance model with no simulated covariance, (c, d) covariance model with simulated covariance. In both instances 120 trees each were assessed in higher and lower true disease prevalence locations. Error bars represent 95% confidence intervals, and dashed line represents perfect agreement between estimated values and actual values.
(TIF)

**S16 Fig. Sensitivity and specificity distribution output using simulated survey datasets (120 trees, good knowledge of both symptoms' sensitivity and specificity, reliable true disease prevalence knowledge).** Probability density of estimated sensitivity and specificity of surveyors using good sensitivity and specificity prior distributions with reliable prior knowledge of true disease prevalence with (a, b) no covariance model with no simulated covariance, (c, d) covariance model with simulated covariance. In both instances 120 trees each were assessed in higher and lower true disease prevalence locations by 25 surveyors. Red crosses represent true surveyor sensitivity and specificity. Solid red line represents the 50th percentile (median), dotted red lines represent the 5th and 95th percentiles of the distributions surveyor sensitivity and specificity values were generated from.
(TIF)

**S17 Fig. The estimated sensitivity and specificity for individual simulated surveyors against their actual sensitivity and specificity values (120 trees, good knowledge of both symptoms' sensitivity and specificity, reliable true disease prevalence knowledge).** Good sensitivity and specificity prior distributions were used with reliable prior knowledge of true disease prevalence with (a, b) no covariance model with no simulated covariance, (c, d) covariance model with simulated covariance. In both instances 120 trees each were assessed in higher and lower true disease prevalence locations. Error bars represent 95% confidence intervals, and dashed line represents perfect agreement between estimated values and actual values.
(TIF)

**S18 Fig. Output errors in relation to the model prior distributions for models with reliable prior knowledge of true disease prevalence.** The maximum error (i.e., difference from the actual value of a simulated surveyor) of a median value estimate for either the sensitivity or specificity against the maximum percentile distance of an actual value of a simulated surveyor from the median of a prior distribution (sensitivity, specificity, site true disease prevalences, or covariance parameters).
(TIF)

**S19 Fig. Sensitivity and specificity distribution output using simulated survey datasets (very good knowledge of one symptom's sensitivity and specificity, true disease prevalence underestimated).** Probability density of estimated sensitivity and specificity of surveyors using poor sensitivity and specificity prior distributions for symptom one, and very good sensitivity and specificity prior distributions for symptom two with misspecified prior knowledge which underestimated the true disease prevalence at survey locations, with 25 surveyors each assessing (a, b) 80, (c, d) 100, or (e, f) 120 trees in each of the higher and lower true disease prevalence locations. Red crosses represent true surveyor sensitivity and specificity. Solid red line represents the 50th percentile (median), dotted red lines represent the 5th and 95th percentiles of the distributions surveyor sensitivity and specificity values were generated from.
(TIF)

**S20 Fig. The estimated sensitivity and specificity for individual simulated surveyors against their actual sensitivity and specificity values (very good knowledge of one symptom's sensitivity and specificity, true disease prevalence underestimated).** Poor sensitivity and specificity prior distributions for symptom one, and very good sensitivity

and specificity prior distributions for symptom two were used with misspecified prior knowledge which underestimated the true disease prevalence at survey locations, with surveyors assessing (a, b) 80, (c, d) 100, or (e, f) 120 trees in each of the higher and lower true disease prevalence locations. Error bars represent 95% confidence intervals, and dashed line represents perfect agreement between estimated values and actual values.
(TIF)

**S21 Fig. Sensitivity and specificity distribution output using simulated survey datasets (good knowledge of both symptoms' sensitivity and specificity, true disease prevalence underestimated).** Probability density of estimated sensitivity and specificity of surveyors using good sensitivity and specificity prior distributions with misspecified prior knowledge which underestimated the true disease prevalence at survey locations, with 25 surveyors each assessing (a, b) 80, (c, d) 100, or (e, f) 120 trees in each of the higher and lower true disease prevalence locations. Red crosses represent true surveyor sensitivity and specificity. Solid red line represents the 50th percentile (median), dotted red lines represent the 5th and 95th percentiles of the distributions surveyor sensitivity and specificity values were generated from.
(TIF)

**S22 Fig. The estimated sensitivity and specificity for individual simulated surveyors against their actual sensitivity and specificity values (good knowledge of both symptoms' sensitivity and specificity, true disease prevalence underestimated).** Good sensitivity and specificity prior distributions with misspecified prior knowledge which underestimated the true disease prevalence at survey locations, with surveyors assessing (a, b) 80, (c, d) 100, or (e, f) 120 trees in each of the higher and lower true disease prevalence locations. Error bars represent 95% confidence intervals, and dashed line represents perfect agreement between estimated values and actual values.
(TIF)

**S23 Fig. Sensitivity and specificity distribution output using simulated survey datasets (poor knowledge of both symptoms' sensitivity and specificity, true disease prevalence overestimated).** Probability density of estimated sensitivity and specificity of surveyors using poor sensitivity and specificity prior distributions with misspecified prior knowledge which underestimated the true disease prevalence at survey locations, with 25 surveyors each assessing (a, b) 80, (c, d) 100, or (e, f) 120 trees in each of the higher and lower true disease prevalence locations. Red crosses represent true surveyor sensitivity and specificity. Solid red line represents the 50th percentile (median), dotted red lines represent the 5th and 95th percentiles of the distributions surveyor sensitivity and specificity values were generated from.
(TIF)

**S24 Fig. The estimated sensitivity and specificity for individual simulated surveyors against their actual sensitivity and specificity values (poor knowledge of both symptoms' sensitivity and specificity, true disease prevalence overestimated).** Poor sensitivity and specificity prior distributions were used with misspecified prior knowledge which underestimated the true disease prevalence at survey locations, with surveyors assessing (a, b) 80, (c, d) 100, or (e, f) 120 trees in each of the higher and lower true disease prevalence locations. Error bars represent 95% confidence intervals, and dashed line represents perfect agreement between estimated values and actual values.
(TIF)

**S25 Fig. Sensitivity and specificity distribution output using simulated survey datasets (very good knowledge of one symptom's sensitivity and specificity, true disease prevalence overestimated).** Probability density of estimated sensitivity and specificity of surveyors using poor sensitivity and specificity prior distributions for symptom one, and very good sensitivity and specificity prior distributions for symptom two with misspecified prior knowledge which overestimated the true disease prevalence at survey locations, with 25 surveyors each assessing (a, b) 80, (c, d) 100, or (e, f) 120 trees in each of the higher and lower true disease prevalence locations. Red crosses represent true surveyor sensitivity and

specificity. Solid red line represents the 50th percentile (median), dotted red lines represent the 5th and 95th percentiles of the distributions surveyor sensitivity and specificity values were generated from.
(TIF)

**S26 Fig. The estimated sensitivity and specificity for individual simulated surveyors against their actual sensitivity and specificity values (very good knowledge of one symptom's sensitivity and specificity, true disease prevalence overestimated).** Poor sensitivity and specificity prior distributions for symptom one, and very good sensitivity and specificity prior distributions for symptom two were used with misspecified prior knowledge which overestimated the true disease prevalence at survey locations, with surveyors assessing (a, b) 80, (c, d) 100, or (e, f) 120 trees in each of the higher and lower true disease prevalence locations. Error bars represent 95% confidence intervals, and dashed line represents perfect agreement between estimated values and actual values.
(TIF)

**S27 Fig. Sensitivity and specificity distribution output using simulated survey datasets (good knowledge of both symptoms' sensitivity and specificity, true disease prevalence overestimated).** Probability density of estimated sensitivity and specificity of surveyors using good sensitivity and specificity prior distributions with misspecified prior knowledge which overestimated the true disease prevalence at survey locations, with 25 surveyors each assessing (a, b) 80, (c, d) 100, or (e, f) 120 trees in each of the higher and lower true disease prevalence locations. Red crosses represent true surveyor sensitivity and specificity. Solid red line represents the 50th percentile (median), dotted red lines represent the 5th and 95th percentiles of the distributions surveyor sensitivity and specificity values were generated from.
(TIF)

**S28 Fig. The estimated sensitivity and specificity for individual simulated surveyors against their actual sensitivity and specificity values (good knowledge of both symptoms' sensitivity and specificity, true disease prevalence overestimated).** Good sensitivity and specificity prior distributions with misspecified prior knowledge which overestimated the true disease prevalence at survey locations, with surveyors assessing (a, b) 80, (c, d) 100, or (e, f) 120 trees in each of the higher and lower true disease prevalence locations. Error bars represent 95% confidence intervals, and dashed line represents perfect agreement between estimated values and actual values.
(TIF)

## Acknowledgments

We would like to thank all involved in the AOD survey days and the National Trust and Royal Parks for allowing workshops to take place on their properties.

## Author contributions

**Conceptualization:** Matt Combes, Nathan Brown, Alexander Mastin, Stephen Parnell.

**Formal analysis:** Matt Combes, Robin N. Thompson, Alexander Mastin, Stephen Parnell.

**Funding acquisition:** Nathan Brown, Peter Crow, Stephen Parnell.

**Investigation:** Nathan Brown, Peter Crow.

**Methodology:** Matt Combes, Nathan Brown, Robin N. Thompson, Alexander Mastin, Stephen Parnell.

**Project administration:** Nathan Brown, Stephen Parnell.

**Supervision:** Stephen Parnell.

**Visualization:** Matt Combes.

**Writing – original draft:** Matt Combes, Stephen Parnell.

**Writing – review & editing:** Matt Combes, Nathan Brown, Robin N. Thompson, Alexander Mastin, Peter Crow, Stephen Parnell.

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
