## [Decision Letter · Decision Letter 0]

28 Apr 2025

PCOMPBIOL-D-25-00441

Unlocking plant health survey data: an approach to quantify the sensitivity and specificity of visual inspections

PLOS Computational Biology

Dear Dr. Combes,

Thank you for submitting your manuscript to PLOS Computational Biology. After careful consideration, we feel that it has merit but does not fully meet PLOS Computational Biology's publication criteria as it currently stands. Therefore, we invite you to submit a revised version of the manuscript that addresses the points raised during the review process.

Please submit your revised manuscript within 30 days Jun 28 2025 11:59PM. If you will need more time than this to complete your revisions, please reply to this message or contact the journal office at ploscompbiol@plos.org. Please include the following items when submitting your revised manuscript:

We look forward to receiving your revised manuscript.

Kind regards,

Nik J. Cunniffe

Academic Editor

PLOS Computational Biology

Benjamin Althouse

Section Editor

PLOS Computational Biology

**Additional Editor Comments :**

I was fortunate enough to receive comments on this paper from four reviewers, all expert on the topic covered. Overall the reviews are positive, particularly Reviewers #1 and #4. Although Reviewer #4 has some minor questions around novelty, given the reference to prior art (comment on L471-473) is from the grey literature, having overlooked this is understandable, although it should be addressed in the revision. And anyway, overall Reviewer #4 is convinced this is potentially an important and interesting contribution. So am I.

However, Reviewer #2 asks for some clarifications on precisely what was done, to allow them to understand whether all assumptions were justified. And Reviewer #3 is - currently at least - somewhat less convinced than the other reviewers. As that reviewer suggests, this might be due to presentation and/or differences of opinion about the precise meaning of gold standard. However, it is also possible that this reviewer has identified technical issues that need to be resolved, and so it is important to respond in some detail. I intend to send the resubmitted version of the m/s - as well as the response to reviews - back to Reviewers #2 and #3, since I feel they should see your responses to their comments and have a chance to comment further.

Finally, about Reviewer #4's request for a user-friendly tool, if this was possible then that would clearly be a very nice contribution. However, given you have already supplied code, if this is not possible then that would not block publication. However, I do have some sympathy with Reviewer #4's contention that there is an interaction here with the description as a "workflow", and if it is impossible to provide something for non-experts to use, you might consider some softening of the nomenclature?

**Journal Requirements:**

2) Thank you for including an Ethics Statement for your study. Please include:

i) The full name(s) of the Institutional Review Board(s) or Ethics Committee(s)

ii) The approval number(s), or a statement that approval was granted by the named board(s).

4) We notice that your supplementary Figures are included in the manuscript file. Please remove them and upload them with the file type 'Supporting Information'. Please ensure that each Supporting Information file has a legend listed in the manuscript after the references list.

Potential Copyright Issues:

i) Please confirm (a) that you are the photographer of 1, or (b) provide written permission from the photographer to publish the photo(s) under our CC BY 4.0 license.

ii) Figure 2. Please (a) provide a direct link to the base layer of the map (i.e., the country or region border shape) and ensure this is also included in the figure legend; and (b) provide a link to the terms of use / license information for the base layer image or shapefile. We cannot publish proprietary or copyrighted maps (e.g. Google Maps, Mapquest) and the terms of use for your map base layer must be compatible with our CC BY 4.0 license.

**Reviewers' comments:**

Reviewer's Responses to Questions

Reviewer #1: This is a very well written manuscript with the data analyzed appropriately and provides novel insights into symptom recognition. The stated objectives of the study to provide a workflow to quantify sensitivity and specificity in the absence of a gold-standard for risk-based surveillance strategies, to understand how sensitivity and specificity differ for symptom types, and the variability in sensitivity and specificity among raters is well-addressed. The authors demonstrate how sensitivity and specificity depend on the symptoms, the individual surveyor, and the survey protocol. This is novel work.

Reviewer #2: The manuscript focuses on the analysis of the sensitivity and specificity of visual inspections conducted by citizen scientists. These metrics were calculated using data provided by an expert assessor, due to the unavailability of standard reference data. Acute Oak Decline was used as a case study to explore how these parameters vary depending on symptom type, individual surveyor, and symptom frequency. Additionally, a Bayesian modelling workflow was developed for parameter estimation even in the absence of an expert evaluator, incorporating uncertainty and prior knowledge. The results show high variability in sensitivity and specificity depending on the factors analysed. The authors suggest that the results can be extrapolated to other diseases with similar symptoms, although they note that adaptations would be required for different pathologies, and that future studies should account for the possibility of an asymptomatic period.

The manuscript is well structured, and the data collection and analysis process is clearly described. The usefulness of the results and the proposed workflow are discussed, along with directions for future research. I agree with the authors that evaluating sensitivity and specificity in visual inspections is an important consideration in the design of surveillance strategies, and this study represents a valuable step forward in this area. I also appreciate that the code and data have been made publicly available. My suggestions and comments are listed below.

Analyses of AOD survey data:

• Please, clarify how sensitivity and specificity were calculated using the expert assessor as a reference.

• Although the analysis is described in general terms, a more detailed specification of the model would be helpful. In particular, the model equation should be provided, and the response variable explicitly stated. Based on the description, it is not entirely clear whether the response variable corresponds to a binary data or to a proportion given an aggregation. Clarifying this point would improve the reproducibility and interpretation of the results.

• Since generalised linear models were used, it is assumed that the individual surveyor was included as a fixed effect. If the aim is to generalise beyond the specific set of surveyors included in the study, it might be more appropriate to model this factor as a random effect. Although a fixed effect specification may be reasonable, especially with a small number of surveyors, it would be desirable to briefly justify this choice.

Methods to quantify sensitivity and specificity in the absence of a ‘gold-standard’ expert dataset:

• Could you please confirm whether “individual” in the context of the multinomial categories refers to each tree assessed by surveyors? This would help ensure we correctly interpret the structure of the model.

• Please, clarify what is meant by 'test' in this context, as the term is not explicitly defined (e.g. “In this context, each test refers to the assessment of a specific symptom by a given surveyor”)

Conclusions:

• Although it is highlighted that the results can be applied to diseases with similar symptoms, it might be helpful to further discuss how the variability observed in sensitivity and specificity could influence the applicability to other diseases.

Reviewer #3: Review of PCOMPBIOL-D-25-00441 Unlocking plant health survey data: an approach to quantify the sensitivity and specificity of visual inspections

This paper asks a relevant question on assessing the quality of citizen scientists as assessors of introduced tree disease. The analysis is sophisticated and the topic is forward looking and I think the topic is important and fits the journal. I enjoyed reading the paper. However, despite having sufficient background in the subject area (invasion ecology and statistical modelling in ecology) and reading the paper twice, I did not fully understand it and I feel unconvinced that the basic tenet of the paper is fulfilled, i.e. that the sensitivity and specificity of visual inspections of disease symptoms by, e.g., citizens, can be quantified without a gold standard. I find this hard to believe. This lack of being convinced may be an error on my side, or perhaps a misinterpretation of the authors’ intent, so in this review I will try and explain what I found not clear and where my skepticism comes from. Maybe I have simply misunderstood some of the language used by the authors and clarifications will address the issues. I hope the comments below will enable the authors to further elaborate and justify their conclusions in a revision.

Detailed comments

25 May be shortened to: “Knowing sensitivity and specificity of visual inspection is critical …”

26-27 “To address this” should refer to a problem, but the problem was not stated.

29 “these trees” at this point the reader cannot understand whether these are the exact same trees or the same species of trees or trees in the same area. Please clarify.

31 Replace “these parameters” by “sensitivity and specificity” for clarity

32 Suggest omitting “rarely available” or put it in comma’s or brackets or use “, which is rarely available”.

34 at this point it is unclear whether “number of symptoms” means the number of different symptoms that were assessed, or the number of leaves or branches with symptoms on a tree.

34 Suggest writing “type of” symptoms rather than “symptoms”

34-36 Here it is unclear whether one merely needs to know which site has relatively low prevalence and which has relatively high prevalence, or one needs to know the values (0.3 and 0.6) of prevalence. Importantly, readers need an intuition why, when using a Bayesian estimation framework, there would not be a need to know the actual prevalence of disease to estimate sensitivity and specificity of citizen observers as long as there are two sites differing in prevalence available with the only knowledge being available is that the sites differ (or perhaps one is higher the other is lower in prevalence). Giving the reader this intuition is critical (at least for me it is). I did not get this intuition from the present version.

51 “these trees” It is not clear whether this means the exact same individual trees or the same species of tree.

54 “these parameters” -> sensitivity and specificity

55 omit “rarely available” and add it at the end “, which is rarely available”

70 is the impact of 5 billion euro per year or is it accumulated (and potentially discounted) over a certain period?

71-76 Suggest to add common names for the disease and the tree species.

77 such -> introduced (or alien or invasive) (for clarity)

80 suggest to omit “strategies”

85/86 suggest to start new paragraph here to improve readability

101-102 This text is clearer than previous text stating an expert had monitored these trees.

102 Suggest to start a new paragraph at “Next”

103 see previous comment on “these parameters”

112, 113 Use full name Acute Oak Decline (Abbreviations not suitable in title). Line 103 is perhaps a proper place to introduce the abbreviation AOD after the full term.

114 Add common names perhaps?

146 Not sure what OS grid reference is. Can you add information or a reference how readers can use this to locate the sites?

149 Since 2019 and 2010

168-170 I suppose you mean that 20 of the 23 volunteers had experience with tree work as well as knowledge of AOD, but I am not entirely sure, given the way it is formulated.

181 I suppose you mean that sensitivity and specificity of the detection of EACH separate symptom was assessed. Please clarify as such so readers are not in doubt.

183-185 Did you make one overarching model in which the type of symptoms was a covariable or did you analyse data separately for each symptom?

185 logit link function

194-196 I cannot get my head around the statement that using Bayesian estimation, the sensitivity and specificity of diagnostic tests can be estimated when two or more tests are applied to the same individuals across two or more populations with different true disease prevalences without knowledge on the actual prevalence in these areas. This is a premise that is fundamental to the paper and I feel it warrants more text to help the reader get an intuition. I am aware that references are provided, but this principle is so important that I think it needs elaboration in the paper itself. And I am doing this review on an intercontinental flight without the benefit of the internet.

197 The way this is written you seem to say that these methods assume independence of sensitivity and specificity, but is that possible at all? Or perhaps you mean independence of the test result for different tests on the same individual, which should be unlikely if the tests are proper? Please clarify.

197-198 The way this is written it seems you do not use this non-independence, but I think you do, don’t you?

200 “We apply these models” Which ones? With or without assumed independence?

202-208 Here a method is elaborated for two tests (i.e. two symptoms). But in the case study you had three symptoms. It would be good to mention in the very beginning that the theoretical case (simulations) is elaborated for two symptoms, not three as in the case study. I only realized this later.

217-222 I am puzzled by reading about separate covariances for disease positives and disease negatives. Usually, we have a covariance between values of two variables, irrespective of the values. Here there are two covariances, one for negative outcomes of two tests and another for positive outcomes. Is this still the common definition of covariance or is it a different thing altogether? Maybe this is better explained in 245-247, but then I am already confused.

238 Do you report the minimum and maximum values?

254-257 This sentence is hard to understand. You are calculating something (covariances) that was assumed in the first place, but you don’t say why. Is this to find out how well you could estimate covariances from the generated data? What were the minimum and maximum values? That is still based on the case study? So, to what extent is the simulation exercise based on the case study?

258-268 This is hard to follow. It might be helpful to provide a mathematical derivation of these metrics, starting at theory that most readers are probably familiar with. It could be in SI if it is unwieldy.

Fig. 3 Caption should state what is given by the curve with the purple area under the curve (priors?). Then the dashed distributions are the assumed parameter distributions for generating data for analysis?

290-291 How close to the case study are we? Are you using actual data on bleeds, cracks and exit holes?

292 Please provide context for your anticipation. This is your expectation about what data might be available in practice? Do you expect that data will be available to estimate priors for prevalence in a high and low prevalence area?

293 prior knowledge: add “on prevalence”

298-299 How do you know the minimum and maximum possible values?

321 “as described previously” might be replaced by an actual list of parameters estimated.

341 Better use full name for title

346 How is the frequency of symptoms defined? Is frequency the same as prevalence? If so, use one term.

346 These are interactions in the glm? Maybe use the term “dependency”?

Fig. 5 what is the reason that the CI is different for surveyors having similar sensitivity? Why is there less certainty on the performance of some surveyors than others? Did they sample fewer trees?

402 But does having such information not qualify as having a (gold) standard? Or is that just a silver or bronze standard? Maybe I misread the article because of misinterpretation of the term “gold standard”. Even if you don’t have exact information on the disease status of each and every tree, as long as you have a distribution (and an average), you can estimate how good a test is (though there is a chance of false positives making up for missed true positives). There is perhaps also a more fundamental issue here. In normal Bayesian inference, we estimate a parameter for which we have a prior. So we estimate a posterior, given the data, to update the prior. But it is still the same variable. Here, the prevalence is uncertain (but a prior is available), and you estimate test results, where some positives may be missed and some negatives may be mislabeled. Then, the result of analysis is another estimate of prevalence, and sensitivity and specificity are then probably (is my understanding) still calculated as if the priors can be used as a “standard”. Is my assumption right or am I wrong? How not-gold is such a standard?

474 hinders

476-477 Rephrase. Now it says” Ability between […] different symptoms. I get your meaning, but reads awkward.

480 Omit “rarely available” Insert “lack of” before “informative prior information”

483 Add “required” after “assessments”

490 It is not obvious how you have used your data to conclude that “particular symptoms will influence optimal survey locations”. Please explain.

495 I’d say “an Agrilus species” not another Agrillus species as EAB is the only Agrilus species that has killed millions of ash trees in the USA.

496 are directly comparable to -> “are similar to” or “resemble” (also in 498). Insert “the” before A. biguttatus

503 abundance is the same as frequency, a word used earlier in the text?

508 Suggest stating: “All citizen observers involved in the study …”

518 species –> parasitoid (damage scores are not a species)

550 The word “envisage” is not so clear. You mean “recommend”? To whom? To achieve what?

Reviewer #4: This is a very well-written and competently executed study on an important topic. It makes a novel contribution to current knowledge and the proposed workflow has potential to be influential in operational biosecurity. Over all, I think it is great, and recommend it for publication with minor revisions.

My only real reservation is that after looking at the supplied code, the touted “workflow” for estimating sensitivity and specificity is not really approachable except by experts. It would be great if they could supply something more user-friendly like a tool, maybe with a structured Excel file for practitioners to enter how many plants were sampled, by who, and what was observed. Then a script would suck that in and spit out the results. This may be asking too much, but it is what I thought of when I saw “workflow”.

I have few other suggestions, except for a criticism that, based on some written statements and the rather Euro-centric citations, the authors seem unaware of a lot of relevant work from Australasia and North America. I have some sympathy with the authors as I am acutely aware that the sheer volume of research is now such that it is impossible to be familiar with it all. But in this case, it renders some of their statements inaccurate, including that they are the first to estimate sensitivity and specificity of visual inspections for plant disease in the absence of a gold standard. More details below…

Line 82: I suggest replacing “usually” with “often” since non-visual sampling methods are the norm for non-pathogen pests.

Line 86 to 88: Contrary to this statement, Cardwell et al. 2018 (https://apsjournals.apsnet.org/doi/10.1094/PHP-06-18-0036-RV) write "There are hundreds of diagnostic plant disease assays used every day in the United States that either are not validated or are validated in an ad hoc way." Does this indicate a difference between Europe and America?

Line 88 to 89: This is not true in Australasia, where the probability of detecting a pest via initial visual inspection has been studied a lot, admittedly with relatively few of those studies being published. Some citeable examples include: Bulman et al. 1999 (http://www.scionresearch.com/__data/assets/pdf_file/0004/17248/NZJFS291_1999_102_115BULMAN.pdf), Froud et al 2008 (https://nzpps.org/_oldsite/books/2008_Surveillance/Surveillance.pdf), Hauser & McCarthy 2009 (http://onlinelibrary.wiley.com/doi/10.1111/j.1461-0248.2009.01323.x/abstract), Mangano et al. 2011 (https://benthamopenarchives.com/abstract.php?ArticleCode=TOENTOJ-5-15), Hester & Cacho 2012 (http://www.tandfonline.com/doi/abs/10.1080/10807039.2012.632307), Hammond et al. 2016 (https://doi.org/10.1016/j.cropro.2015.10.004), Caley et al. 2020 (http://link.springer.com/10.1007/s10340-019-01115-7).

Line 203: It is not clear to me what is meant by “test one” and “test two”. It feels like a little more explanation is needed to set up here.

Line 278: I suggest pointing out that the priors are shown in purple.

Line 471 to 473: The authors will be disappointed to learn that their study is not the first to derive quantitative estimates of the sensitivity and specificity of visual inspection for symptoms in plant health in the absence of a gold-standard reference dataset: see Chapter 4 in Froud et al 2022 (https://www.knowledgeauckland.org.nz/media/2398/tr2022-08-2021waitakere-ranges-kauri-population-health-survey06chapter-4.pdf). However, the simulation exploration, workflow and R code of the current study are very welcome enhancements.

Line 510 to 511: See Mangano et al. referenced above.

**Have the authors made all data and (if applicable) computational code underlying the findings in their manuscript fully available?**

Reviewer #1: Yes

Reviewer #2: Yes

Reviewer #3: Yes

Reviewer #4: Yes

PLOS authors have the option to publish the peer review history of their article (what does this mean? ). If published, this will include your full peer review and any attached files.

**Do you want your identity to be public for this peer review?** For information about this choice, including consent withdrawal, please see our Privacy Policy .

Reviewer #1: No

Reviewer #2: No

Reviewer #3: No

Reviewer #4: **Yes: ** John Kean

**Figure resubmission:**
---

## [Decision Letter · Decision Letter 1]

4 Sep 2025

PCOMPBIOL-D-25-00441R1

Unlocking plant health survey data: an approach to quantify the sensitivity and specificity of visual inspections

PLOS Computational Biology

Dear Dr. Combes,

Thank you for submitting your manuscript to PLOS Computational Biology. After careful consideration, we feel that it has merit but does not fully meet PLOS Computational Biology's publication criteria as it currently stands. Therefore, we invite you to submit a revised version of the manuscript that addresses the points raised during the review process.

==

Thank you for taking such care over the revision. The paper is in very good shape, and the previously critical reviewer is much happier. They suggest a number of small possible elaborations; please act on these as you please, and resubmit. I will make a final decision on that submission without another round of review.

==

Please submit your revised manuscript within 30 days Nov 04 2025 11:59PM. If you will need more time than this to complete your revisions, please reply to this message or contact the journal office at ploscompbiol@plos.org. Please include the following items when submitting your revised manuscript:

We look forward to receiving your revised manuscript.

Kind regards,

Nik J. Cunniffe

Academic Editor

PLOS Computational Biology

Benjamin Althouse

Section Editor

PLOS Computational Biology

**Reviewers' comments:**

Reviewer's Responses to Questions

**Comments to the Authors:**

Reviewer #3: Review of PCOMPBIOL-D-25-00441_R1 Unlocking plant health survey data: an approach to quantify the sensitivity and specificity of visual inspections

Authors have made an effective revision that results in a much clearer manuscript (at least for this reader). Upon reading the revision, I still found a few things less clear, and these are pointed out below. I think it is easy to fix and it is at the discretion of the authors as far as I am concerned. It is very interesting work, and I feel it will be welcomed in the community, though I suspect many people will find it difficult to understand. Hence, there remains a need to try and be as clear and explicit as possible to avoid readers getting lost.

33 “using simulated data” Readers may wonder “simulated data for WHAT”? It might help if this could be stated explicitly.

97 “this makes it” -> “it is”

118-120 These sentences are very vague. Readers may really be helped if you state in words what you do (conceptually) before getting into the details (as you do in the methods). This would help them getting their bearings before it gets more mathematical (and many readers will skip). You want that those readers can still correctly interpret the figures. This would be helped by stating clearly that you assess sensitivity and specificity of individual assessors and give readers an intuition how that is achieved.

130 Spelling: buprestid

Fig 1a caption. There is only a single bleed on the photo. Revise text accordingly.

194 Spelling: symptom

208, 209 I suppose you need to have “sensitivities” and “specificities” (plural); otherwise there can be no covariances. Or does the use of singular work in English? You are native speakers, I am not.

302 I found myself looking for the very good prior knowledge of sensitivity and specificity (in purple colour) and did not find it. Can you add “(not shown in the figure)”?

325 I feel it is a little more standard to state that surveyor observations were “generated” from distributions rather than simulated. I make this remark because the word “simulated” is used in different contexts (understandably), e.g. also where you simulate sensitivity and specificity of individual assessors. In the latter context, I would find the word “simulate” more appropriate. But maybe that is an idiosyncratic preference that the authors don’t share. Anyhow, If the terms used could be fine-tuned to the actual things you do it might be helpful to readers.

325 “using 0.3” -> “using a prevalence of 0.3” (I feel it would be much clearer)

391 add “and” at the end of the line

392 “values” -> “values of sensitivity and specificity” would be clearer (more explicit)

393 Here you might use “expert data as a gold standard” or just “expert data as a standard”?

413 I suppose you mean here the sensitivity and specificity of EACH assessor, rather than the distribution of sensitivity and specificity in the WHOLE population of assessors. Please make it more explicit.

417 Did you mean here that the GENERATED specificity value of the assessor was not supported by the prior? More clarity is needed what is the meaning of “simulated”.

424 Maybe add “for same assessors” before the left bracket (

424 What is meant by “poor knowledge both symptoms”? Please write out your meaning for clarity. Isn’t “poor knowledge both symptoms contradictory to “reliable knowledge on disease prevalence”?

432 “With this reliable prior knowledge” does not follow up from the preceding text (above Fig. 6

Fig. 7 caption: same comment as for Fig. 6 caption.

527 “which are more likely to be most effective spread wider across the landscape” Not clear. Check language/grammar

551-552 I think it was stated in the beginning of the paper that some of the citizen surveyors had been professional surveyors in the past. It may contradict this statement.

599 The methodology is quite complex, and as noted by another reviewer, the code may not be particularly easy to use for non-experts, and I suppose that is an understatement. Hence, consider dropping “practical”.

Fig. 8 caption: see comments on captions of Figs 6 and 7. “Poor sensitivity and specificity prior distributions” -> “uninformative prior distributions of sensitivity and specificity”.

**Have the authors made all data and (if applicable) computational code underlying the findings in their manuscript fully available?**

Reviewer #3: Yes

PLOS authors have the option to publish the peer review history of their article (what does this mean? ). If published, this will include your full peer review and any attached files.

**Do you want your identity to be public for this peer review?** For information about this choice, including consent withdrawal, please see our Privacy Policy .

Reviewer #3: No

**Figure resubmission:**
---

## [Editor Report · Decision Letter 2]

7 Nov 2025

Dear Dr Combes,

We are pleased to inform you that your manuscript 'Unlocking plant health survey data: an approach to quantify the sensitivity and specificity of visual inspections' has been provisionally accepted for publication in PLOS Computational Biology.

Best regards,

Nik J. Cunniffe

Academic Editor

PLOS Computational Biology

Benjamin Althouse

Section Editor

PLOS Computational Biology

---

## [Editor Report · Acceptance letter]

PCOMPBIOL-D-25-00441R2

Unlocking plant health survey data: an approach to quantify the sensitivity and specificity of visual inspections

Dear Dr Combes,

I am pleased to inform you that your manuscript has been formally accepted for publication in PLOS Computational Biology. Your manuscript is now with our production department and you will be notified of the publication date in due course.

With kind regards,

Judit Kozma
